# Calibration and performance evaluation of PM$_{2.5}$ and NO$_2$ air quality sensors for environmental epidemiology

Miriam Chacón-Mateos[1,a], Héctor García-Salamero[1], Bernd Laquai[1], and Ulrich Vogt[1]

[1]University of Stuttgart, Institute of Combustion and Power Plant Technology, Department of Flue Gas Cleaning and Air Quality Control, Stuttgart, 70569, Germany

[a]now at: German Aerospace Center, Institute of Combustion Technology, Stuttgart, 70569, Germany

**Correspondence**: Miriam Chacón-Mateos (miriam.chacon-mateos@ifk.uni-stuttgart.de)

**Abstract.** Over the past few decades, the study and the use of air quality sensors have significantly increased, leading to a wealth of experience and a deeper understanding of their strengths and limitations. This study aimed to develop and evaluate a methodology for PM$_{2.5}$ and NO$_2$ sensors to enhance sensor accuracy to a level suitable for epidemiological studies, where ensuring data quality is paramount. The performance evaluation of indoor and outdoor sensors was carried out during the co-location phase with reference-equivalent instruments (RIs), by calculating the relative expanded uncertainties (REUs) stated in the EU Air Quality Directive 2008/50/EC and the recently published EU Directive 2024/2881, target diagrams and common error metrics, before the deployment of the air quality sensor systems (AQSSs) in the houses of patients suffering from chronic obstructive pulmonary disease (COPD) or asthma in Stuttgart (Germany). Regression and machine learning models for sensor calibration were tested during the co-location. Moreover, an original methodology was designed and evaluated to validate the sensor data during the deployment in the houses of the participants. The study found that indoor sensor calibration using artificially generated NO$_2$ and aerosols does not ensure model transferability, emphasizing the need for training data that matches the intended deployment environment in terms of real patterns of concentration, particle composition and environmental conditions. Moreover, the impact of the aggregation time (1, 5, 10 and 15 min) on the performance of the calibration models was evaluated for NO$_2$ sensors. Integrating metadata such as activity logs, window status, and data from official monitoring stations, as well as NO$_2$ measurements with diffusion tubes proved to be helpful for data validation and interpretation during the sensor deployment in the houses of the participants.

**Keywords** Low-cost sensors; indoor air; outdoor air; PM$_{2.5}$; NO$_2$; Epidemiological studies; Measurement uncertainty

## 1 Introduction

The World Health Organization (WHO) updated its global air quality guidelines in September 2021. The new air quality recommendations proposed by the WHO resulted from the findings based on recent epidemiological studies. The increase in evidence on the adverse health effects of air pollution has been possible thanks to the advances in technology for air pollution

monitoring and personal exposure (WHO, 2021). A major air pollutant is particulate matter (PM), especially the fine fraction

$PM_{2.5}$, which can cause respiratory and cardiovascular diseases, reproductive and central nervous system dysfunctions, and cancer (Manisalidis et al., 2020). In a meta-analysis, Braithwaite et al. (2019) also found statistically significant associations between long-term $PM_{2.5}$ exposure and mental illnesses such as depression and anxiety. Another air pollutant of special interest is $NO_2$, which has been associated with higher morbidity for vulnerable groups such as asthma and chronic obstructive pulmonary disease (COPD) patients (Hoffmann et al., 2022). Moreover, a recent review paper has shown that both short- and

long-term exposure to $PM_{2.5}$ or $NO_2$ adjusted for $NO_2$ and $PM_{2.5}$, respectively, revealed a synergistic effect appearing as higher mortality from respiratory diseases (Mainka and Żak, 2022).

Exposure measurements are carried out using direct or indirect approaches. The direct approaches measure the exposure levels by using personal passive sampling devices (Piechocki-Minguy et al., 2006; Shirdel et al., 2019; Samon et al., 2022) or mobile monitors (Rea et al., 2001; Koehler et al., 2019) that must be worn by the person during the campaign. In recent years more

studies have deployed air quality sensors allowing multi-pollutant exposure assessment (Piedrahita et al., 2014; Chatzidiakou et al., 2020; Novak et al., 2021). This methodology is considered the most accurate estimate of a person's 'true' exposure. However, this type of personal exposure assessment is only adequate for short-term exposure (Steinle et al., 2013). The main challenges of these studies are the complexity of the data integration including the time-activity-location profiles (Chatzidiakou et al., 2022), and the measurement uncertainty due to the position of the sampling inlet, which may be largely affected by the

perihuman/personal cloud effect (Licina et al., 2017; Pantelic et al., 2020). In theory, the sampling inlet should be placed close to the breathing zone, but this is in reality not always feasible, especially for multi-pollutant devices (Yun and Licina, 2023; Bendl et al., 2023). Additional factors, such as vibrations, static electricity (Shirdel et al., 2019) and movement (e.g. isokinetic sampling of PM cannot not be guaranteed), have also an influence on the accuracy of the measurement. Moreover, other external factors like the accuracy of the GPS signal, the accelerometer, etc. may be crucial to characterize the true exposure.

The indirect approaches measure air quality at fixed monitoring sites or are based on modelling (Goldman et al., 2012; Beloconi and Vounatsou, 2020; Huang et al., 2021) which can also integrate satellite data (Hang et al., 2022). Among the indirect approaches, some studies rely on outdoor measurements at fixed-site monitoring stations (Harré et al., 1997; Meng et al., 2013). This has been the cause of exposure misclassification in the past (Shaw et al., 2018), as outdoor monitoring stations fail to capture the real concentrations in the different microenvironments an individual is exposed to (Krause, 2021). Moreover,

strong correlations among the ambient pollutants can lead to biased health effect estimates due to confounding (Sarnat et al., 2001). Other indirect approaches are based on static measurements in the most visited microenvironments of the participants (Scott Downen et al., 2022). The main advantage of this methodology is the lower effort required of the participant which allows longer measurement periods, making it the ideal candidate for long-term exposure assessment (Steinle et al., 2013).

In this context, some studies have evaluated the use of stationary air quality sensors for environmental epidemiology

(Morawska et al., 2018; Patton et al., 2022; Bi et al., 2024; Zuidema et al., 2024). Zuidema et al. (2021) evaluated the field calibration based on series of stepwise multiple linear regression calibration models of a low-cost sensor network for multiple gaseous pollutants. They reported the performance achieved using the CV-RMSE and the CV-$R^2$ as well as the limitations of

the approach to, for instance, detect the drift of the sensors during deployment or the difficulty to measure low pollution levels. They also discussed about the competing interests forcing the compromise between duration of co-location in order to achieve better calibration (training data) and the deployment for epidemiological purposes.

The use of air quality sensors for environmental epidemiology has many advantages, for instance, the decrease in the bias of exposure estimations when compared with fixed outdoor monitoring stations (Chatzidiakou et al., 2019). Another benefit of using sensors is the possibility of increasing the number of participants with the same fixed budget, which helps to ensure adequate statistical power of the study. Moreover, sensors allow time resolutions in the order of seconds, making possible the exposure assessment on movement and the correlation of pollution patterns with personal behaviour when this information also exists (Jerrett et al., 2017; Novak et al., 2022). However, although the high temporal resolution offered by sensors is valuable for capturing dynamic exposures, it also increases instrumental noise, which directly impacts measurement uncertainty (Schmitz et al., 2025).

On the other side, some characteristics of the sensors have kept them away from applications where high accuracy is required. One of them is the influence of meteorological conditions such as temperature (T) and relative humidity (RH) and cross-sensitivities in the sensor signal (Samad et al., 2020; Venkatraman Jagatha et al., 2021; Zamora et al., 2022). That makes the calibration of the sensors more complex than traditional monitoring devices, as the calibration algorithms should account also for those influences, and that limits the transferability of the calibration models when moving the sensor to a different location (Zauli-Sajani et al., 2021; Diez et al., 2024). Another parameter that affects the sensor accuracy for long-term measurements is the signal drift caused by the sensor degradation (Tancev, 2021; deSouza et al., 2023). Last but not least, the unit-to-unit variability poses a challenge when it comes to calibrating many units at the same time, as is the case for epidemiological studies (Gäbel et al., 2022).

Some recent studies have shown that the above-mentioned concerns can be overcome and that getting highly personalized air pollution exposure outweighs the measurement uncertainty of the air quality sensors. The AIRLESS study (Effects of AIR pollution on cardiopuLmonary disEaSe in urban and peri-urban reSidents in Beijing) demonstrated that sensing technologies can revolutionise health studies and address scientific, health and policy questions in a way that has not been possible before (Krause, 2021). The results of the AIRLESS project have been well-documented (Chatzidiakou et al., 2020; Krause, 2021) and are a prove of the potential use of sensing technologies for epidemiological studies in very different environments, i.e. high- and middle-income countries like London (Evangelopoulos et al., 2021) and Beijing (Han et al., 2020; Han et al., 2021) but also low-income countries like Kenya (Krause, 2021).

Recent literature demonstrates that stationary indoor air quality measurements with low-cost sensors are widespread, but calibration approaches and durations vary considerably (Anastasiou et al., 2022; Soja et al., 2023; Tryner et al., 2021; Rathbone et al., 2025). Rose et al. (2024) investigated and apportioned the sources of indoor PM at school classrooms using the OPC-N3 (Alphasense, UK). The calibration was carried out using linear regression using data from the co-location with a RI during the exposure to indoor air for 48 h using a time resolution of 1 min. Good agreement for $PM_{2.5}$ (r > 0.85) was reported, without the need of a further correction to account for hygroscopic growth as the RH was below 60 %.

Suriano and Penza (2022) tested the performance of Alphasense series B4 sensors for CO, $NO_2$ and $O_3$ during a one-week co-location experiment in a living room using a sampling rate of 2 min. The models tested for calibration were multiple linear regression (MLR), random forest regressor (RFR), artificial neural networks (ANN) and support vector regressor (SVR), and the input parameters used were the working electrode (WE), the auxiliary electrode (AE), the T and the RH or also the net difference WE – AE, including also the T and RH. They proved that the $NO_2$ measurements were in good agreement ($R^2 >$ 0.7, 8.4<MAE< 12 ppb, 10.6 < RMSE< 16.3 ppb) if calibrated through MLR, RF and ANN, having the best results when using separately the sensor electrode signals as inputs. Note that in both studies the co-location was short, as the pumps of the RI are too noisy to keep the instrument longer periods in such indoor environments.

As shown in the aforementioned examples from the literature, it is common practice to report sensor accuracy primarily through metrics such as $R^2$ or Pearson correlation coefficient, with some studies including additional statistics like MAE or RMSE. However, these statistical evaluations alone may not be sufficient for specific purposes as well as for stakeholders such as environmental agencies, who work with expensive instrumentation that undergo rigorous calibration and continuous performance assessments throughout their operational lifespan (Flores et al., 2012; Flores et al., 2013). Therefore, to build greater confidence in air quality sensor data, more comprehensive validation protocols and calibration procedures are essential. Figure 1 shows the link between epidemiological studies, the WHO and the European Directives for air quality. The epidemiological studies are the prove of causality between air pollutant exposure and health effects, and they are reviewed by the WHO to recommend the limit values which are the guidance to set the air quality regulations. In the European Union, the EU Directive 2008/50/EC and the new Directive 2024/2881 specify the short- and long-term limit values, as well as the Data Quality Objectives (DQOs) that the measurements must meet for ambient air quality assessment, depending on the type of measurement (fixed, indicative or objective estimation). At this moment there is no DQOs for indoor air quality assessment. However, in this work we have evaluated the sensor data for the indicative and objective estimation DQOs set in the directives for both indoor and outdoor measurements.

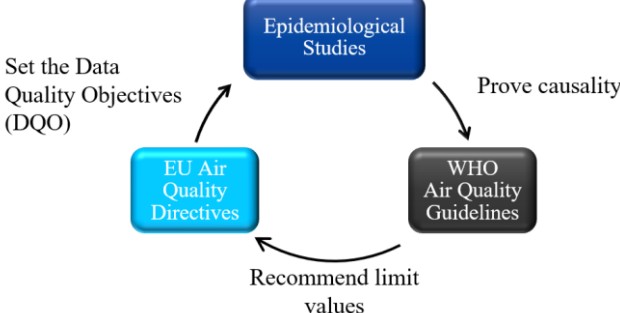

**Figure 1.** Interconnection between epidemiological studies, the WHO air quality guidelines, and the DQOs established in the EU Air Quality Directives.

This work aims to evaluate the performance of $NO_2$ and $PM_{2.5}$ sensors for their use in health research. We present an approach to calibrate the sensors based on co-location with RIs and assess the reliability of the calibration before and during deployment. The sensors were deployed in the houses of seven COPD and asthma patients. The measurements were conducted in two microenvironments per participant representing the outdoor and the indoor levels of exposure for one month. The MLR and three machine learning (ML) models (RFR, SVR, and ANN) have been evaluated for indoor and outdoor calibration of $NO_2$ sensors and different averaging times (1, 5, 10 and 15 min). A univariate linear regression (ULR) calibration was investigated to correct the $PM_{2.5}$ sensor measurements. The outdoor $PM_{2.5}$ sensor included a thermal drying inlet. The performance evaluation has been carried out using common error metrics, REUs according to the European data quality objectives (DQOs), and target diagrams. Finally, we discuss the capabilities as well as the limitations of the proposed methodology.

## 2 Methodology

### 2.1 Participant recruitment and study protocol

The participants consisted of seven patients suffering from COPD or asthma. All the participants lived in Stuttgart (Germany) (see Fig. S1 in the Supplement) and agreed to perform the measurements in their homes for 30 days. One participant agreed to install two outdoor AQSSs instead of one, to compare street-side and garden-side concentrations. For this participant, the measuring campaign was reduced to 19 days.

The study protocol was evaluated and approved by the Ethics Committee of the Medical Association of the State of Baden-Württemberg (reference number F-2019-105) and by the data protection officer of the University of Stuttgart. Before the beginning of the measurements, participants were informed about the study and requested to provide written consent. The participants are referred to by a patient identification number from P1 to P7. An environmental questionnaire in the German language was designed to characterize the living area, the house, and the habits, and was completed prior to the measurements with the help of a worker of the University of Stuttgart. Participants also completed a spirometry test, a health survey on their symptoms, and a logbook documenting hourly indoor activities, window status and presence at home. This information collected from each participant has been further analysed in Chacón-Mateos et al. (2024).

At the end of the measurements, we asked the participants for written feedback. Participants who started the study before March 2020 received the study indications at their homes. However, those who started the study after the COVID-19 outbreak performed the interview by phone, and the contact between the participants and the university staff was kept to a minimum. A detailed description of the data collected and the further analysis to determine the feasibility of using the developed AQSSs and methodology for exposure assessment and indoor source apportionment can be read in Chacón-Mateos et al. (2024).

### 2.2 Indoor and outdoor air quality sensor systems

Two different AQSSs for indoor and outdoor measurements were designed for this study (see Fig. 2), each one containing one electrochemical sensor for $NO_2$ (Alphasense, UK, model B43F), and one optical particle counter for $PM_{2.5}$ (Alphasense, UK,

model OPC-R1). The sensor selection was based on our own tests of different sensors in the laboratory. Another important

factor that was considered was the price, being 150 Euro the maximum possible price per sensor. Additionally, a T and RH sensor was included (IST AG, Switzerland, model HTY221). The microcontroller Arduino UNO was used to control and save the data every two seconds on an SD card. Both AQSSs had a passive ventilation system. During the deployment, participants did not have access to the data in order to avoid behavioural changes.

As an outdoor AQSS must be weather resistant, we selected an enclosure made of glass fibre-reinforced polyester with the

160 following dimensions: 200×300×150 mm. For the indoor AQSS, a polypropylene box with the dimensions 240×195×112 mm was chosen. The cost of the materials amounted to a total of 540 and 460 Euro for the outdoor and indoor AQSSs, respectively. To counteract the effect of the high RH in the PM sensor readings, a low-cost dryer was designed for the outdoor PM sensor. The main advantage of using a low-cost dryer is that it allows the use of the same calibration models independently of the location of the PM sensor. Other techniques based on the κ-Köhler theory or machine learning have shown incorrect results

when moving the sensor to another location, as the particle composition may differ from the one in the co-located site (Di Antonio et al., 2018; deSouza et al., 2022). The dryer consists of a 50 cm brass tube with a resistive wire wound around its surface. The wire is heated when the RH is higher than 70 % using 12 V and 10 W. The T is controlled by using the internal T sensor of the OPC-R1. A detailed description and evaluation of the low-cost dryer can be read in Chacón-Mateos et al. (2022).

The indoor AQSSs (B01, B02 and B04) were installed in the participants' living rooms, as this space was identified as the primary area for their daily activities. The exact placement within the living room was determined by the proximity to a power outlet and the availability of suitable space, with devices most commonly positioned on a table or TV stand. Outdoor AQSSs (B03, B05, B06 and B08) were installed in a variety of locations, including hanging from balconies, placed on window sills, or positioned on terrace floors, with placement always dependent on the availability of a power socket. A summary of the

information collected in the environmental questionnaire about the neighbourhood, the building as well as the home environment, including the type of windows, possible pollutant sources indoors and outdoors, can be read in Chacón-Mateos et al. (2024).

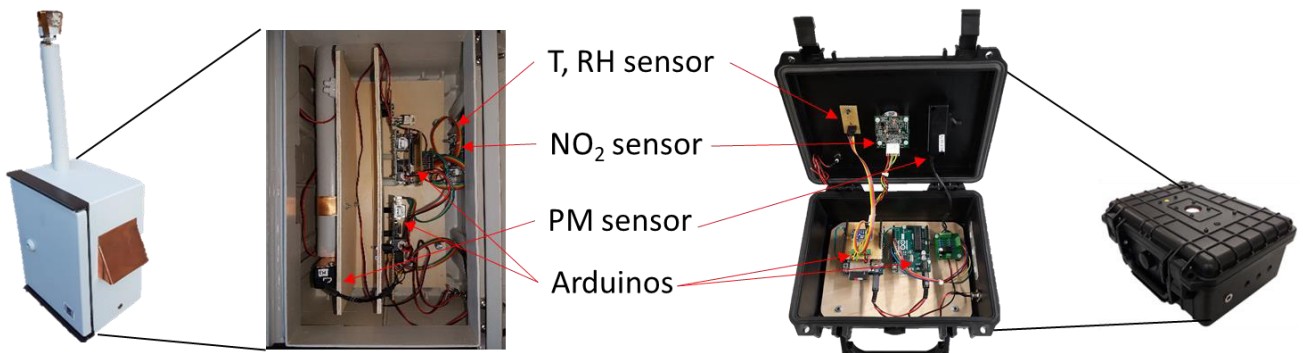

**Figure 2.** Designed AQSSs for outdoor (left) and indoor (right) measurements (Chacón-Mateos et al., 2024).

 **2.3 Quality assurance**

The measurements in the houses of the patients took place in Stuttgart region (Germany) between 20 December 2019 and 28 May 2020. Figure S1 shows the approximate locations of the participants' homes, the governmental outdoor air quality monitoring stations in Stuttgart and the monitoring stations of the University of Stuttgart. The co-location of the indoor and outdoor AQSSs took place discontinuously starting on 7 November 2019 and finishing on 5 June 2020 and was done the weeks before the individual deployment in the patient´s houses or immediately after it. A general overview of the measuring campaign showing the periods where the co-location and deployment of the AQSSs took place can be seen in Fig. S2. In the following subsections, a detailed description of the methodology used to verify and assess the quality of the data before and after the deployment in the homes of the patients as well as the calibration procedures are described.

**2.3.1 Sensor co-location before deployment**

The co-location for both indoor and outdoor AQSSs were conducted in distinct locations to replicate real-world environmental conditions as closely as possible. Likewise, the methodology was tailored to address the specific conditions encountered in indoor as well as outdoor environments. The main objective was to cover the maximum range of possible concentrations, T and RH that could be found later in the indoor and outdoor locations. A summary of the different procedures can be seen in Table 1.

Before deployment, the $NO_2$ sensors for indoor measurements were co-located in the laboratory for a minimum of seven days and a maximum of 34 days, depending on the availability of the AQSSs. A chemiluminescence device (MLU, Austria, model 200A) was used as RI for $NO_2$. Due to the low $NO_2$ concentrations measured in the laboratory, it was necessary to generate higher concentrations using a Gas Phase Titration system (GPT) (Ecotech, Australia, model Serinus Cal 3000). For this purpose, the indoor AQSSs were placed inside a sealed box made of inert glass with gas supply connections. The dimensions of the box were as follows: 310 × 525 × 375 mm. The sensors were exposed to the following pyramid of $NO_2$ concentrations: 0 - 50 - 100 - 50 - 0 ppb. Each stage was maintained for 3 hours and the pyramid was repeated at least twice in different days. The changes in the T and RH were forced using an infrared lamp close to the calibration box and an air humidifier, respectively. Moreover, natural changes in the room conditions were simulated by opening and closing the windows of the laboratory.

**Table 1.** Co-location methodology of the NO$_2$ and PM$_{2.5}$ sensors.

| Pollutant | Indoor AQSSs | Outdoor AQSSs |
|---|---|---|
| NO$_2$ | Co-location in the laboratory:<br>- Low concentrations (< 10 ppb): indoor air.<br>- High concentrations (up to 180 µg m$^{-3}$): artificial generated NO$_2$.<br>- Changes in T using an infrared lamp.<br>- Changes in the RH by manually opening and closing the windows and using an air humidifier. | Co-location at Hauptstätter Street monitoring station. |
| PM$_{2.5}$ | Co-location:<br>- In the laboratory (real exposure to indoor air).<br>- High concentrations (up to 150 µg m$^{-3}$): calibration aerosol in particle chamber. | Co-location at Hauptstätter Street monitoring station. |

The co-location of PM$_{2.5}$ sensors was performed in a particle chamber. High particle concentrations (up to 150 µg m$^{-3}$) with a peak concentration at an aerodynamic diameter of less than 3 µm were dispersed using an aerosol generator and liquid paraffin. To account for potential particle losses caused by electrostatic forces from the plastic enclosure, the entire indoor AQSS was placed inside the chamber. The RI was a light-scattering device (Grimm, Germany, model 1.108). The experiments in the particle chamber took one hour and were repeated twice. More information about this co-location setup can be read in Laquai

and Saur (2017). After the co-location in the particle chamber, the sensors were installed in the laboratory for a minimum of 2 days and a maximum of 27 days to expose the sensors to real PM indoors.

The outdoor AQSSs were co-located seven to 34 days in the hotspot monitoring station at Hauptstätter Street (48°45´55.8936´´ N, 9°10´12.9396´´ E), that belongs to the University of Stuttgart. The average co-location time was 15 days. The advantage of performing the co-location in a hotspot station is that the maximum concentrations expected in the city are covered. However,

low-concentrations are unusual to happen there and that may cause a lack of low-concentrations for the training data. As RI for NO$_2$, the model 405 nm from the company 2B Technologies (USA) was used. An EDM 180 from the company Grimm GmbH (Germany) was used as an RI for the PM measurements. The RI for NO$_2$ was calibrated once a month and the measurements of the Grimm EDM 180 were corrected against gravimetric measurements at the beginning of the campaign.

During the measurement campaign and after having analysed the first results, we decided to experiment with a new calibration

strategy: for patient P7 an outdoor box (B03-P7) calibrated with the data from the co-location in the Hauptstätter Street monitoring station was used for indoor air quality measurements. The reason for that was the high deviation of the indoor NO$_2$ concentrations modelled by the support vector regressor (SVR) and random forest regressor (RFR) models when compared to the results of the measurements carried out with diffusion tubes located in the same place during the deployment in the house of the patients (see Section 3.2.1).

 **2.3.2 Data validation during deployment**

Due to the data reliability problems that air sensors have, it is vital to be able to identify if the AQSSs are working properly during the deployment in the houses of the patients. In an ideal case, having an RI co-located would be the best option. However, this is usually not possible for epidemiological studies with a lot of participants. For that reason, we present here a methodology that can be used in epidemiological studies having a high number of participants. This approach has been

summarized in Table 2.

**Table 2.** Validation of the $NO_2$ and $PM_{2.5}$ sensors during deployment.

| Pollutant | Indoor AQSSs | Outdoor AQSSs |
|---|---|---|
| $NO_2$ | Comparison with diffusion tubes (quantitative). | Comparison with diffusion tubes (quantitative). Comparison with outdoor air quality monitoring stations less than 6 km apart (qualitative). |
| $PM_{2.5}$ | Identification of possible sources of peak concentrations using the activity log (qualitative). | Comparison with outdoor air quality monitoring stations less than 6 km apart (qualitative). |

To have a reference $NO_2$ concentration value in the houses, $NO_2$ passive samplers (diffusion tubes) from the company Passam

(Switzerland) were attached to the indoor and outdoor AQSSs to perform discontinuous measurements. In this technique, $NO_2$ is absorbed in a metal mesh that has been treated with triethanolamine (DIN EN 16339). After 14 days of exposure time, the diffusion tubes were collected and analysed in the laboratory as described in VDI 2453 Part 1 (1990). The agreement or disagreement of the sensor data with the diffusion tubes was quantified by comparing the values of $NO_2$ measured with the diffusion tubes during 14 days to the average of the continuous sensor data using different calibration models during those 14

days. For patients P2 and P4, only one period was collected of 14 and 19 days, respectively.

Additionally, the data of the four outdoor air quality monitoring stations available in Stuttgart as well as the data of the monitoring station of the University of Stuttgart in Hauptstätter Street was also collected to qualitatively compare their $NO_2$ and the $PM_{2.5}$ trends with the data of the outdoor AQSSs during deployment in the houses of the patients. The air distances between the closest and the furthest monitoring station and the houses of the patients was 0.6 and 6 km, respectively (see Fig.

S1). Moreover, in order to ensure the quality of the measurements carried out with the diffusion tubes, we co-located three diffusion tubes (triple determination) in the monitoring station of the University of Stuttgart and changed them every 14 days. Due to the lack of passive samplers for $PM_{2.5}$, the indoor $PM_{2.5}$ concentrations could only be validated using the activity logbook, by checking whether peak concentrations coincided with activities likely to generate particulate matter (e.g., cooking, cleaning), or by analysing window status (open/closed) and temperature variations.

**2.3.3 Calibration procedures**

In this section, the calibration procedures used for $PM_{2.5}$ and $NO_2$ sensors are described. For PM sensors the univariate linear regression (ULR) shown in Eq. 1 was used,

$$PM_{2.5,corrected} = \beta_0 + \beta_1 PM_{2.5,raw} \tag{1}$$

where $\beta_0$ is the calibration constant and $\beta_1$ the calibration factor of the linear fitting between the PM$_{2.5}$ concentrations of the sensor and the RI. The use of a low-cost dryer prevents the outdoor PM sensor readings from the influence of hygroscopic
growth of PM when the RH is higher than 70 %. The indoor PM sensor was also calibrated using ULR and it did not include the low-cost dryer as RH higher than 70 % indoors was not expected. During the deployment, we measured indoor RH between 18 to 58 %.

For NO$_2$ sensors, different parametric and non-parametric models were investigated to take into account the influence of the RH and the T in the sensor signal: MLR, RFR, SVR and ANN. These models have been already investigated to correct the
data of air quality sensors with promising results (Esposito et al., 2016; Topalović et al., 2019; Zimmerman et al., 2018; Bigi et al., 2018) but literature about how these models perform when the sensor is transferred to a new location is scarce.

The explanatory variables (also called features in ML models) used for all the models were data of the WE and AE, and the T and RH of the HYT221 sensor. The MLR model shown in Eq. 2 is applied to correct the NO$_2$ sensor data. In Eq. 2, $\alpha_0$ is the intercept and $\alpha_n$ are the coefficients that applied to each explanatory variable.

$$NO_{2,corrected} = \alpha_0 + \alpha_1 WE + \alpha_2 AE + \alpha_3 T + \alpha_4 RH \tag{2}$$

The RFR is an ML algorithm based on ensembles of decision trees (Breiman, 2001). The main characteristics are that it randomizes both the selection of the data points used to build the trees and the explanatory variables at each node to determine the split. Thus, leading to each decision tree being built on a slightly different dataset with a different subset of features (Müller and Guido, 2017). During prediction, the RFR calculates the average of the predicted values from all the decision trees, resulting in a more accurate prediction than a single decision tree. The RFR is known for its ability to handle noisy and complex
data while reducing overfitting and improving model performance.

The SVR models come originally from support vector machine algorithms, which are usually used for classification purposes (Boser et al., 1992). In SVR, instead of trying to minimize the residuals between the predicted values and the actual values using the conventional sum of the squared residuals of a linear fitting, the goal is to find a margin that includes as many data points as possible within a certain distance, also called epsilon ($\varepsilon$), from the predicted values. To achieve that, a hyperplane in
a high-dimensional feature space, i.e. a function, must be found, so that the threshold distance of the $\varepsilon$-tube between the hyperplane and the support vectors is maximized while the errors of the predicted values are minimized. The support vectors are the data points that lie either on the edge of the $\varepsilon$-tube or violate the margin constraints (Awad and Khanna, 2015). This model is very robust in handling outliers.

The ANN is an ML algorithm inspired by the connections among the cells of the nervous system (McCulloch and Pitts, 1943).
In this model, the training data containing the explanatory variables are inserted as input nodes in the network. This input is used in the first step, called forward propagation, to estimate the value of the parameters (biases and weights). These parameters connect the neurons in the hidden layer/s using the selected nonlinear function (so-called activation function) so that a first prediction of the output node, which is in this case the NO$_2$ concentration, can be estimated. As the output from the forward propagation may not be correct, in the second step, the so-called backpropagation, the biases and weights are optimized to

minimize the residual sum of squares between the observed values (NO₂ concentration of the RI) and the predicted values using gradient descent. In order to avoid wrong predictions caused by local minimums, a parameter called learning rate ($\alpha$) should be as small as possible. Note that the smaller the learning rate, the longer the computational time so an optimum must be found (Bishop, 2006; Awad and Khanna, 2015).

The hyperparameter tuning for the ML models was carried out in Python using the *RandomizedGridSearchCV* optimizer provided by the Scikit-learn library. Additionally, Keras and TensorFlow libraries for ANN models were used. In order to avoid overfitting, a 5-fold cross-validation was used. Some of the preliminary hyperparameter values were based on the literature (Wei et al., 2020; Spinelle et al., 2015; Pedregosa et al., 2011) whereas others were manually tested by means of observing how the learning curves react (Géron, 2019). The grid of parameters for each model is shown in Table S1-S3 in the Supplement. Among the whole calibration dataset, 75 % of the data was used for training and the other 25 % for testing. Both datasets were randomly selected. The hyperparameters were tuned for each sensor individually. All the ML models were built using the Scikit-learn library in Python. A total of 217 simulations were run, 96 % of which were completed in less than 15 minutes on a single 2.50 GHz Intel i7-6500U CPU.

### 2.3.4 Data processing

Firstly, in order to identify and remove outliers, data cleaning was carried out using an unsupervised learning algorithm, the Density-Based Spatial Clustering of Applications and Noise (DBSCAN) (Ester et al., 1996), prior to the training of the calibration models. The warm-up period of NO₂ sensors was observed to range from four hours up to three days and was manually removed after visual inspection of the data.

For PM₂.₅ sensors, the data for calibration of the indoor sensors were averaged every 1 min whereas the data of outdoor sensors were averaged every 30 min. In the case of the calibration of the NO₂ sensors, we evaluated the impact of the averaging time in the model performance by using 1-, 5-, 10- and 15-min averages for both training and testing datasets. Note that the NO₂ sensor signal exhibited significant noise, making necessary to balance the number of training data points with effective noise reduction in order to optimise model performance. During the deployment in the houses of the patients, hourly and daily averages were used for the analysis.

For ANN and SVR models, the data of the explanatory variables were normalised from 0 to 1 using Eq. 3,

$$X_N = \frac{X_i - X_{min}}{X_{max} - X_{min}} \tag{3}$$

where $X_N$ is the normalized value, $X_i$ is the feature value ($i$) to be normalized, and $X_{min}$ and $X_{max}$ are the minimum and maximum values of the feature, respectively. After the prediction, the results were transformed back to the real values.

### 2.3.5 Performance evaluation

Following the recommendation of the CEN/TS 17660-1:2021 and the CEN/TS 17660-2:2024, the REU has been calculated to determine whether the sensor data fulfil the DQOs as defined in the Directive 2008/50/EC. On November 24, 2024, the EU

Directive 2024/2881 was published, establishing stricter limit values to be achieved by January 2030. The new directive also specifies in Annex V new DQOs for indicative measurements (I.M.) and objective estimation (O.E.) that the Member States shall comply by 11 December 2026. Therefore, the inclusion of new DQOs is intended as forward-looking exercise.

The CEN/TS 17660-1 (2021) and CEN/TS 17660-2 (2024) provides a classification that is consistent with the requirements of DQOs defined in the Directive 2008/50/EC. Sensors fulfilling the DQO required for indicative measurements belong to class 1 whereas sensors in class 2 fulfil the DQO for objective estimations. A third class, which is less strict and is not formally associated with the Directive, has also been defined. Class 3 is not object of study of this work, as it is not formally linked to a binding DQO.

In Table 3, the DQOs of shor-term $NO_2$ and $PM_{2.5}$ measurements for both Directives, the 2008/50/EC and 2024/2881 are presented. More information about how to calculate the REU can be read in the Supplement. As shown in Table 3, the DQO of the objective estimation for hourly $NO_2$ values has changed from 75 % in Directive 2008/50/EC to 80 % in Directive 2024/2881 whereas the DQO for daily $PM_{2.5}$ mean concentrations has changed from 100 % in Directive 2008/50/EC to 85 % in Directive 2024/2881. For indicative measurements, only the DQO of daily mean concentrations of $PM_{2.5}$ has been re-defined from 50 to 35 %.

Another aspect that should be noted is the average time. Note that the short-term DQOs were conceived for hourly and daily averages for $NO_2$ and $PM_{2.5}$, respectively. For epidemiological studies, however, especially those using portable monitors, 24-h average or even 1-h average may be insufficient, as detecting short-term pollution peaks requires higher temporal resolutions. Moreover, longer co-location periods are not always possible during the exposure assessment campaigns and consequently, the use of a 1-hour average can decrease considerably the available data to train the calibration models and reduce the range of T and RH, as well as the pollution concentration range used. Therefore, in this work, we present the REUs of the $NO_2$ models for different averaging times, that is, 1, 5, 10, and 15 min and thus, an evaluation of the REUs at the limit value is not applicable. Similarly, co-location measurements of indoor $PM_{2.5}$ sensors in a particle chamber with high particle concentrations lasted an average of 2 to 3 hours. Therefore, the uncertainties were calculated for a 1-min average. For outdoor $PM_{2.5}$ sensors where more data points are available, a 30-min average was used so that neither REU for $PM_{2.5}$ measurements for indoor or outdoor are applicable in the region of the limit values.

**Table 3.** DQOs specified as the largest REU for short-term concentrations (Directive 2024/2881, 2024; Directive 2008/50/EC, 2008).

| Air pollutant | DQO I.M. | | DQO O.E. | |
|---|---|---|---|---|
| | 2008/50/EC | 2024/2881 | 2008/50/EC | 2024/2881 |
| $NO_2$ (1 h) | 25 % | 25 % | 75 % | 80 %[a] |
| $PM_{2.5}$ (24 h[b]) | 50 % | 35 % | 100 % | 85 %[c] |

[a] Calculated as the maximum ratio (3.2) over the uncertainty of indicative measurements (see Annex V of EU Directive 2024/2881).

[b] The EU Directives do not include uncertainty for $PM_{2.5}$ hourly values.

[c] According to Annex V of EU Directive 2024/2881: "The uncertainty of objective estimation shall not exceed the uncertainty for indicative measurements by more than the applicable maximum ratio and shall not exceed 85 %".

The results of the PM$_{2.5}$ and NO$_2$ sensors were also evaluated using target diagrams. A target diagram is built using the CRMSE and the MBE of the testing set as the x-axis and y-axis, respectively, both normalised by the standard deviations of the RI ($\sigma_{ref}$). As the values of CRMSE are always positive, the model predictions are plotted in the left quadrants if their standard deviation is lower than the standard deviation of the RI (Zimmerman et al., 2018). The outermost circle of the diagram

corresponds to the performance criteria, set as 1, whereas the inner circle represents the performance goal which has been defined for this study as 0.5, that is, 50 % more stringent (Jolliff et al., 2009; Bagkis et al., 2021). This threshold is an exploratory criterion adopted specifically for the purposes of this study. The performance of the model is better the closer the attained performance score is to the target diagram's origin (Thunis et al., 2012).

Finally, various goodness-of-fit indexes were used to assess the performance of the models including root-mean-square error

(RMSE), centred root-mean-square error (CRMSE), mean bias error (MBE), mean absolute error (MAE), the coefficient of determination ($R^2$), Person correlation coefficient (r), model efficiency (MEF) and fractional bias (FB). The respective formulas and ideal values are summarized in Table S4 of the Supplement. By presenting both conventional performance metrics and more robust diagnostic tools, we aim to enable a broader comparison with other studies as the REUs and target diagram are still scarcely used in the performance evaluation of AQSSs.

## 3 Results

### 3.1 Sensor data validation before deployment

### 3.1.1 Relative expanded uncertainty

The REU of the testing data for the indoor and outdoor PM$_{2.5}$ sensors before the deployment in the houses of the patients can

be seen in Fig. 3. The DQOs of the EU Directive 2008/50/EC and the new EU Directive 2024/2881 for both objective estimation and indicative measurements of PM$_{2.5}$ are also indicated. As shown in Fig. 3 (a), the unit-to-unit variability of indoor PM$_{2.5}$ sensors is significant. Specifically, the PM$_{2.5}$ sensor in B04-P3 meets the DQO for indicative measurements up to 2 µg m$^{-3}$ and 3 µg m$^{-3}$ under Directives 2008/50/EC and 2024/2881, respectively. In contrast, the PM$_{2.5}$ sensor in B01-P4 meets the DQO for objective estimation only for the Directive 2008/50/EC and concentrations higher than approximately

36 µg m$^{-3}$. Three out of six indoor sensors fulfil the DQO for objective estimation set in the Directives 2008/50/EC and 2024/2881 at 12 µg m$^{-3}$ and 14 µg m$^{-3}$, respectively, and meet the DQO for indicative measurements for PM$_{2.5}$ concentration higher than 24 µg m$^{-3}$ and 35 µg m$^{-3}$ for the same directives respectively.

As can be observed in Fig. 3 (b), the unit-to-unit variability of outdoor calibrated sensors is less pronounced, with some sensors reaching the DQO for indicative measurements for concentrations higher than 5 to 6 µg m$^{-3}$ (B06-P4, B06-P7_end) for both

Directives. Four out of nine calibrated sensors fail to fulfil the DQO for indicative measurements of the new Directive

2024/2881 in contrast to only two that do not achieve the DQO for indicative measurements contemplated in the Directive 2008/50/EC. For the latter Directive, most sensors reach the mentioned DQO at concentrations higher than 16 µg m⁻³.

Similar to the indoor AQSSs, the results for outdoor sensors present data from different testing datasets for the same AQSS. For instance, the AQSS B05 was used by two patients (P2 and P4) and therefore calibrated twice before each deployment. The AQSS B03 was used in the houses of three patients but calibrated four times, including an additional co-location period after the deployment in the house of patient P7. In contrast to indoor calibrated sensors, outdoor sensors exhibit generally consistent REU across different deployments, as observed by the overlapping points. This consistency suggests that the calibration method may influence the REU, possibly because the aerosol (liquid paraffin) used in the particle chamber for calibration does not have the same composition as the urban dust. The OPC-R1 sensor has been designed for ambient aerosol monitoring, assuming a refractive index of 1.5+i0, and a density of 1.65 g/ml for the calculation of the PM mass concentration. Additional details regarding the calibration conditions, the $PM_{2.5}$ concentration range and the calibration coefficients can be read in Table S5 in the Supplement.

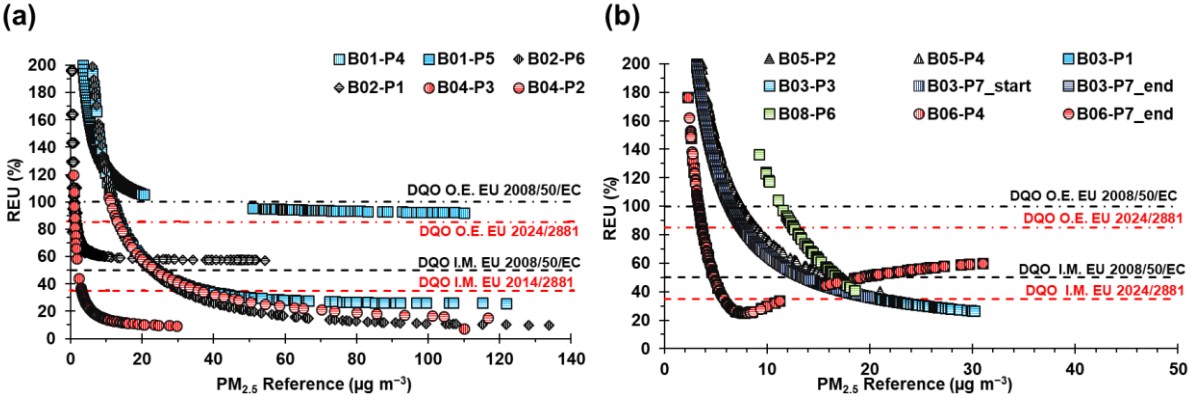

**Figure 3.** REU for (a) indoor and (b) outdoor $PM_{2.5}$ sensors against reference concentration. The coloured symbols are different AQSSs which were deployed in the house of different patients (B0X-PX) and therefore some AQSSs were through more than one calibration phase. B03 was calibrated before the deployment in the house of patient 7 (B03-P7-start) and after the deployment (B03-P7-end). The dashed lines indicate the DQOs for indicative measurements while the dash-dot lines represent the DQOs for objective estimation (black for EU Directive 2008/50/EC and red for EU Directive 2024/2881).

Examples to illustrate the REUs of indoor and outdoor $NO_2$ sensors are shown in Fig. 4, which contains the results of the tested calibration models (MLR, SVR, RFR, and ANN) as well as the influence of the averaging time of the training dataset, for 1, 5, 10, and 15 min on the REU. The DQOs of both Directives 2008/50/EC and 2024/2881 for objective estimation and indicative measurements of $NO_2$ are also indicated. Note that both directives have the same DQO for indicative measurements (25 %). The y-axis has been limited to 110 % so that the difference among the models can be distinguished. In Figures S3 and S4 the diagrams for all the other indoor and outdoor AQSSs are shown, respectively. Additionally, Table 4 and 5 show the

concentration in ppb at which the DQO for indicative measurements (25 %) is accomplished for the outdoor and indoor sensors, respectively.

In general, the coarser the averaging time used for training the data, the lower the REU. However, the longer the averaging time, the smaller the dynamic range of the input variables, which can lead to higher uncertainties due to data extrapolation. Thus, an optimum averaging time shall be used. In our study, we found 10-minute averaging time a good compromise between training the models with enough data points and reaching the DQO for indicative measurements at an average of 23 ppb for the outdoor $NO_2$ sensors. In the Fig. S5 of the Supplement, the number of data points for the training of the $NO_2$ calibration models for the different time resolutions is shown for the indoor and outdoor sensors. A detailed study about the effect of eleven temporal resolutions (between 10 s and 6 h) in the performance of $NO_2$ sensors can be read in Schmitz et al. (2025).

For the sensors calibrated in indoor conditions, SVR and RFR seems to perform better than ANN and MLR. The MLR trained using data averaged 1 min performs in most of the cases the worst. This could be due to the signal noise, not only from the sensor but also from the data of the RI used for the training. Results show that the DQO for indicative measurements (25 %) is achieved with a 10- or 15-min average and $NO_2$ concentrations larger than about approximately 5 - 22 ppb for indoor and 10 - 25 ppb for outdoor AQSSs. The lower REUs that are achieved during the calibration of AQSSs in indoor conditions may be due to the controlled conditions, as the $NO_2$ gas was given stepwise and kept constant for 3.5 hours, as well as the controlled changes of the T and the RH. This lack of variability in the calibration data resulted in low sum of residuals (RSS) triggered by model overfitting. Other authors have also observed better results when the sensors are calibrated in control conditions as compared to outdoor calibrations but they fail later during the field deployment (Castell et al., 2017). This creates the challenge of calibrating indoor AQSSs for a wide range of $NO_2$ concentrations and meteorological parameters without causing model overfitting.

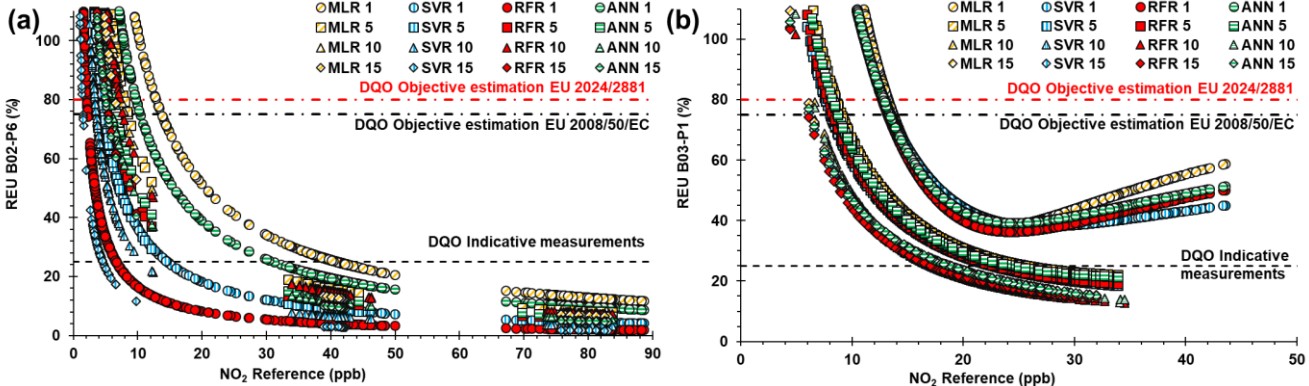

**Figure 4.** Example of REU for (a) indoor and (b) outdoor $NO_2$ sensors for the tested models (in different colours MLR, SVR, RFR and ANN) at different averaging times (in different symbols 1 min, 5 min, 10 min and 15 min) against reference concentrations. The dashed line indicates the DQO for indicative measurements while the dash-dot lines represent the DQOs for objective estimation (black for EU Directive 2008/50/EC and red for EU Directive 2024/2881). The DQO for short-term indicative measurements is the same in both Directives.

**Table 4.** Concentration in ppb at which the DQO for indicative measurements (25 %) is accomplished for the outdoor calibration.

| Averaging time | Model | B03-P1 | B03-P3 | B03-P7* | B05-P4 | B06-P4 | B06-P7 | B08-P6 |
|---|---|---|---|---|---|---|---|---|
| 1 min | MLR | N.A. | N.A. | N.A. | N.A. | N.A. | N.A. | N.A. |
| | SVR | N.A. | N.A. | N.A. | N.A. | N.A. | N.A. | N.A. |
| | RFR | N.A. | N.A. | N.A. | N.A. | N.A. | N.A. | N.A. |
| | ANN | N.A. | N.A. | N.A. | N.A. | N.A. | N.A. | N.A. |
| 5 min | MLR | 26 | N.A. | 27 | N.A. | 39 | N.A. | N.A. |
| | SVR | 23 | 27 | N.A. | N.A. | 34 | 23 | 21 |
| | RFR | 22 | 28 | 21 | N.A. | 33 | 20 | 21 |
| | ANN | 24 | 40 | 38 | N.A. | 39 | N.A. | 28 |
| 10 min | MLR | 17 | N.A. | N.A. | N.A. | 29 | N.A. | 28 |
| | SVR | 17 | 23 | 19 | N.A. | 29 | N.A. | 14 |
| | RFR | 17 | 22 | 19 | N.A. | 24 | 33 | 17 |
| | ANN | 17 | 25 | 32 | N.A. | 28 | 38 | 19 |
| 15 min | MLR | 18 | N.A. | N.A. | N.A. | 19 | N.A. | 24 |
| | SVR | 18 | N.A. | 26 | 35 | 19 | 21 | 11 |
| | RFR | 17 | 29 | N.A. | 30 | 20 | N.A. | 13 |
| | ANN | 18 | 44 | N.A. | N.A. | 18 | N.A. | 19 |

N.A.: not accomplished.

*B03-P7 is an outdoor AQSS used for indoor measurements as part of an experiment to test the outdoor calibration methodology for indoor measurements.

**Table 5.** Concentration in ppb at which the DQO for indicative measurements (25 %) is accomplished for the indoor calibration.

| Averaging time | Model | B01-P4 | B01-P5 | B02-P1 | B02-P6 | B04-P2 | B04-P3 |
|---|---|---|---|---|---|---|---|
| **1 min** | MLR | 8 | 11 | N.A. | 42 | 9 | 11 |
| | SVR | 3 | 2 | N.A. | 15 | 2 | 8 |
| | RFR | 1 | - | 40 | 7 | - | 3 |
| | ANN | 5 | 5 | N.A. | 31 | 5 | 7 |
| **5 min** | MLR | 7 | 10 | N.A. | 27 | 8 | 9 |
| | SVR | 1 | 14 | N.A. | 15 | - | 6 |
| | RFR | 1 | - | 60 | 21 | - | 2 |
| | ANN | 4 | 4 | N.A. | 22 | 4 | 5 |
| **10 min** | MLR | 6 | 10 | N.A. | 26 | 8 | 8 |
| | SVR | 1 | 1 | N.A. | 11 | 1 | 3 |
| | RFR | 1 | - | 39 | 25 | - | 1 |
| | ANN | 3 | 4 | N.A. | 21 | 4 | 5 |
| **15 min** | MLR | 7 | 9 | N.A. | 22 | 7 | 7 |
| | SVR | 1 | 4 | N.A. | 4 | 3 | 3 |
| | RFR | 1 | - | N.A. | 20 | - | 3 |
| | ANN | 3 | 4 | N.A. | 19 | 4 | 4 |

N.A.: not accomplished.

The cells marked with (-) do not have a value for the REU as $U_{field}(y_i)$ cannot be calculated with Eq. S6 in the Supplement due to the negative value of $u_s^2(y_i)$ (Eq. S1). This is caused due to the extremely low RSS. Near-zero RSS are an indicator of the overfitting of the RFR in the indoor calibration models.


### 3.1.2 Target diagrams

The target diagrams for the testing data of the indoor and outdoor PM$_{2.5}$ sensors are shown in Fig. 5. Two main results can be inferred from these diagrams: (i) Different outcomes are obtained with the same sensor for each calibration period, as indicated by the symbols with the same form and colour and (ii) the results of indoor PM$_{2.5}$ sensors remain within the unit circumference, being most of them even within the inner circle, which is 50 % more stringent. In contrast, four out of seven outdoor PM$_{2.5}$ sensors do not perform well enough to reach the inner circle, and most of them remain outside the unit circumference. The differences between the indoor and the outdoor sensors' performances can be attributed to the same factors discussed in Section 3.1.1. Other researchers have obtained similar results, with PM$_{2.5}$ sensors falling within and without the target circle without specific patterns (Borrego et al., 2016). The question of whether the prototype of the dryer unit helped to improve the

performance of the PM$_{2.5}$ sensors of the outdoor AQSSs may arise after analysing this outcome. In Chacón-Mateos et al. (2022) the weaknesses and strengths of the thermal dryer used for this study were discussed in detail. In that study, it was concluded that the dryer was causing an excess of heating and therefore an underestimation of PM$_{2.5}$ concentrations compared to the RI. In this regard, we have developed a new prototype to keep the air temperature inside the dryer at less than 40 °C.

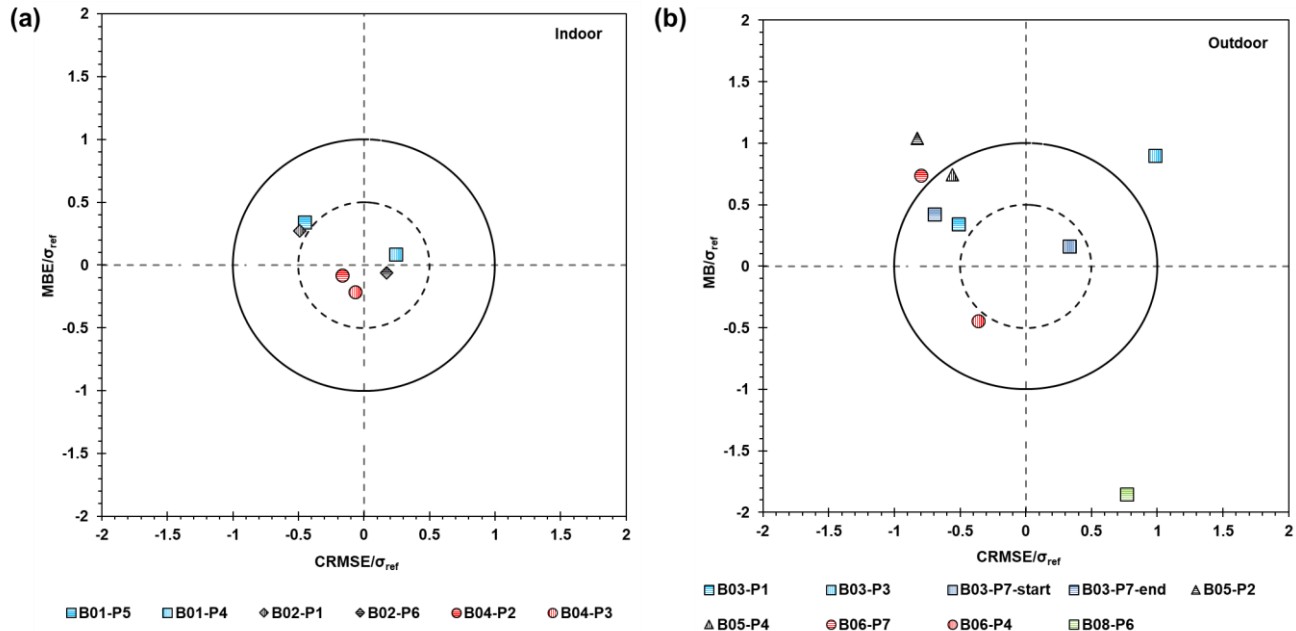


**Figure 5.** Target diagrams for (a) indoor and (b) outdoor PM$_{2.5}$ sensors. The coloured symbols are different AQSSs which were later deployed in the house of different patients (B0X-PX) and therefore some AQSSs were through more than one calibration phase. B03 was calibrated before the deployment in the house of patient 7 (B03-P7-start) and after the deployment (B03-P7-end).

Figure 6 illustrates two examples of target diagrams for the tested models for indoor and outdoor NO$_2$ sensors. The remaining results for indoor and outdoor NO$_2$ sensors are available in Figures S6 and S7 respectively. All the indoor NO$_2$ sensors fall within the performance goal (± 0.5) independently of the average time and the model used, indicating high accuracy (low mean bias or systematic error) and high precision (low CRMSE or random error) for all the models.

The models for correcting NO$_2$ sensor readings outdoors show more discrepancies among the models and averaging times.
Models trained using 1-min averaging time show the worst performance, followed by the 5-min average. For most of the models, the results of the target diagrams for 1- and 5-min averages do not reach the performance target (± 0.5). Higher averaging periods like 10 or 15 min usually reach the inner circle. In terms of models, SVR and RFR tend to outperform MLR and ANN achieving higher accuracy and precision. In all the cases, the results are situated on the left side, indicating that the standard deviation of the sensors was lower than the standard deviation of the RI. This may indicate a systematic
underestimation of the actual variability by the calibration models.

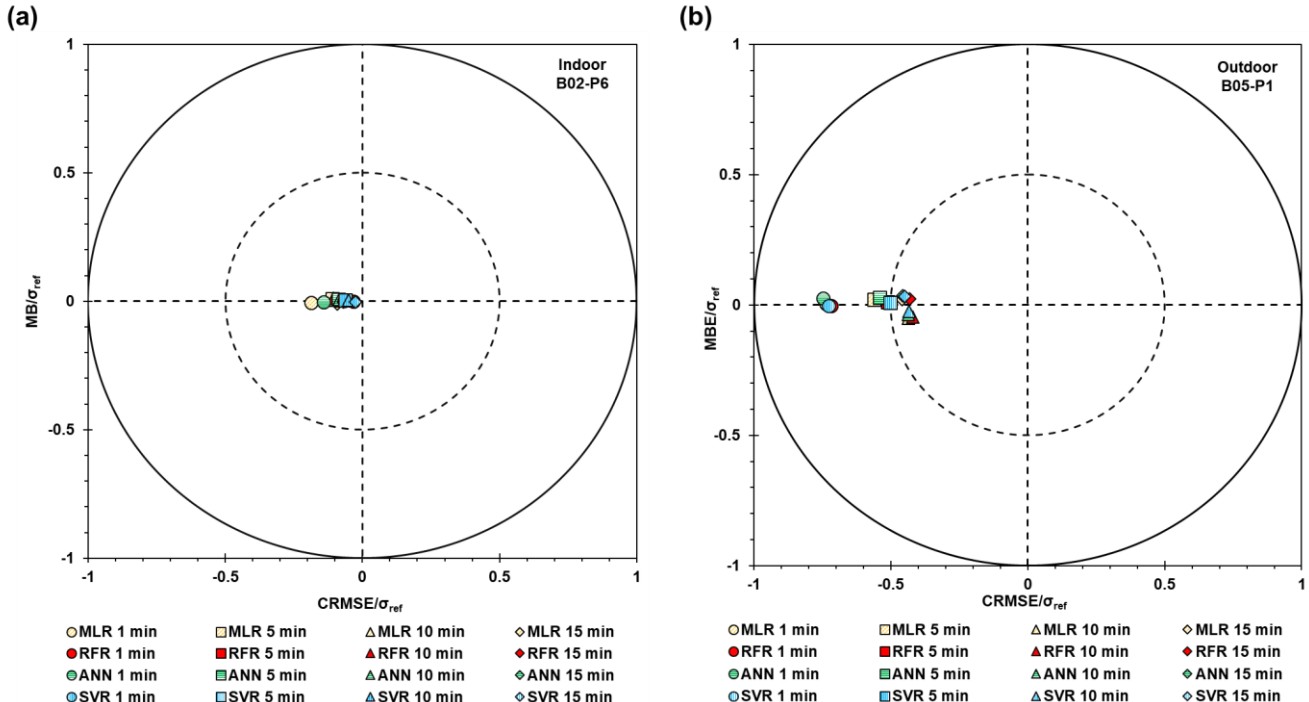

**Figure 6.** Example of target diagrams for (a) indoor and (b) outdoor NO$_2$ sensors for the tested models (in different colours MLR, SVR, RFR and ANN) and different averaging times (in different symbols 1 min, 5 min, 10 min and 15 min).

### 3.1.3 Performance metrics

Figure 7 presents the statistical results for various metrics (orthogonal slope and intercept, model efficiency, MAE, and Pearson correlation coefficient) of the models tested for indoor and outdoor NO$_2$ sensors at different averaging times. Consistent with previous findings, the indoor models outperform the outdoor models, likely due to the more controlled laboratory conditions. Notably, the model efficiency for all indoor models is nearly 1, indicating an almost perfect match to the RI data. When comparing different time aggregations, it is evident that higher aggregation intervals result in the orthogonal slope approaching

1 and the orthogonal intercept approaching 0 for all the tested models. This is attributed to the reduction in sensor noise and increased data stability with higher time aggregation. However, when comparing the MEF for 10- to 15-minute time aggregations, no improvement is observed; instead, there is a decrease in performance across all models. This decline is likely due to the excessive reduction in the number of training data points, with approximately 35 % fewer data points (see Fig. S5). This trend is also observed in the MAE, which decreases from an average of 10 ppb across all models with 1-min averaging

time to 5 ppb using 10- and 15-min averaging times for outdoor NO$_2$ sensors. The improvement in the indoor NO$_2$ sensors is less notable. The Pearson correlation coefficient shows an improvement between 1-min and 5-min averaging time but remains stable thereafter for both indoor and outdoor sensors. In general, MLR shows the worst performance across the tested models. SVR and RFR exhibit the best performance, closely followed by ANN.

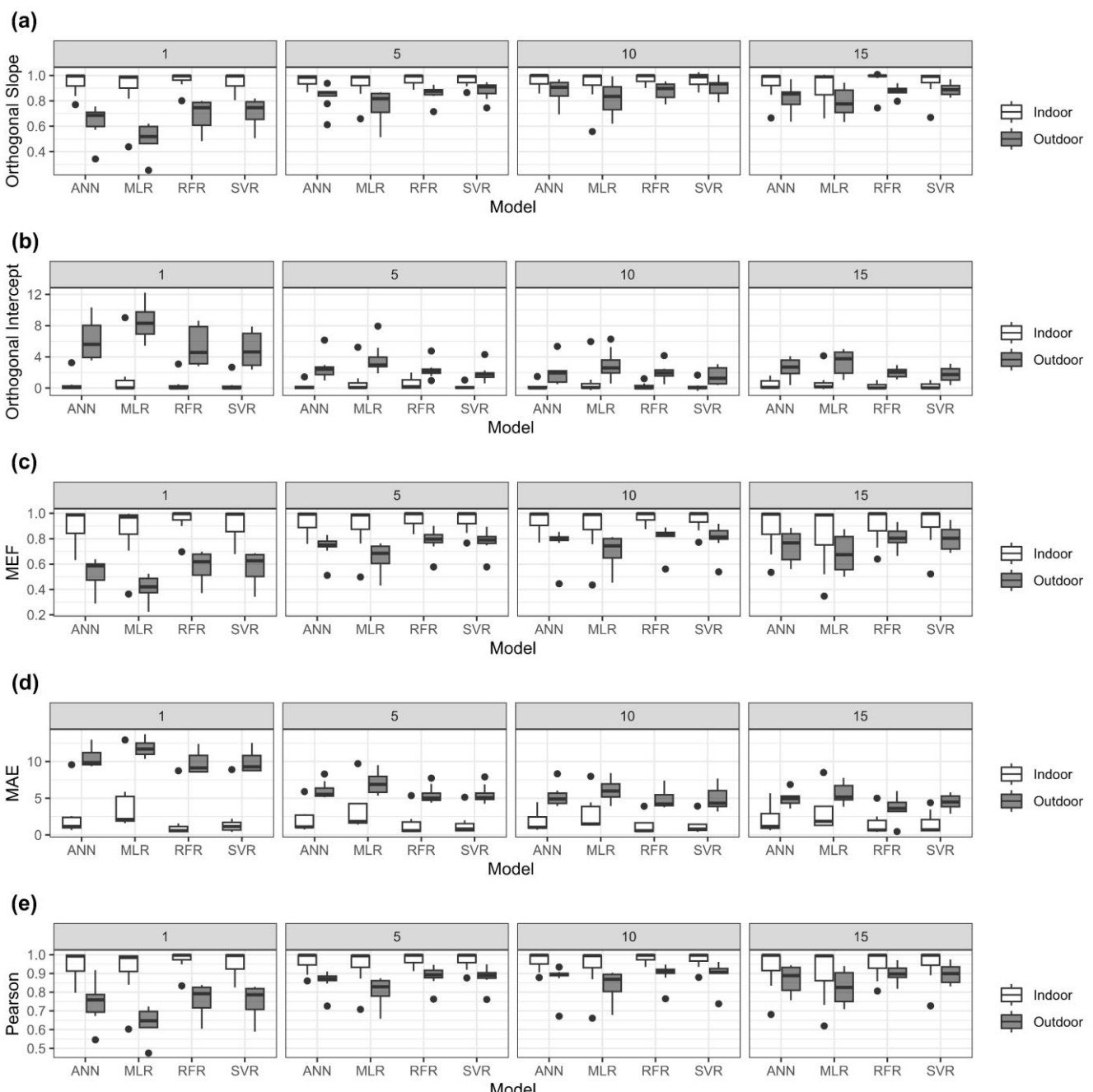

**Figure 7.** Boxplots of various performance evaluation metrics: (a) orthogonal slope, (b) orthogonal intercept (in ppb), (c) model efficiency (MEF), (d) MAE (in ppb) and (e) Pearson correlation coefficient, for different tested models (ANN, MLR, RFR and SVR) for the different time aggregations (1, 5, 10 and 15 min) applied to the testing data for indoor and outdoor $NO_2$ sensors.

Figure 8 presents the performance evaluation metrics for the indoor and outdoor PM$_{2.5}$ sensors. The calibration factor ($\beta_1$) and calibration constant ($\beta_0$) for the indoor sensors are closer to 1 and 0, respectively, compared to the outdoor sensors. Notably, almost all the sensors exhibit a calibration constant greater than zero ($\beta_0 > 0$). This constant deviation, or displacement error, may be attributed to the different limits of detection of the OPC-R1 (0.35 µm) compared to 0.30 µm of the RI. As mentioned in Section 2.3.1, the indoor sensors were calibrated using an aerosol generator and liquid paraffin. However, these particles do not accurately represent the heterogeneity of the particles present in the indoor air. This discrepancy likely explains why the indoor sensors perform better across most metrics except for the MAE, as higher concentrations (median 124 µg m$^{-3}$) were generated during the calibration. In contrast, the highest median PM$_{2.5}$ concentration measured during the outdoor calibration is 35 µg m$^{-3}$. Overall, the calibrated indoor and outdoor sensors exhibit a median FB of less than 0.3, which is within the acceptable limits, and Pearson correlation coefficients of more than 0.75.

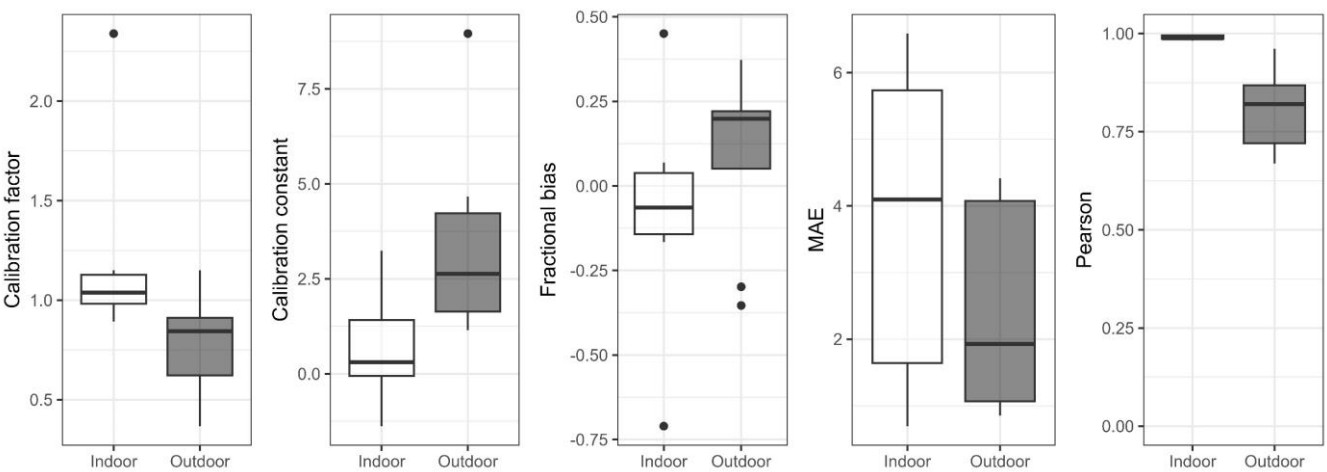

**Figure 8.** Boxplots of various performance evaluation metrics: (a) calibration factor, (b) calibration constant (in µg m$^{-3}$), (c) fractional bias, (d) MAE (in µg m$^{-3}$) and (e) Pearson correlation coefficient, for indoor and outdoor PM$_{2.5}$ sensors.

## 3.2 Sensor data validation during deployment

### 3.2.1 Comparison with the NO$_2$ measurements of the diffusion tubes

Figure 9 presents the results of the discontinuous NO$_2$ measurements using diffusion tubes for the indoor and outdoor microenvironments during the deployment in the houses of the patients, compared to the results of the tested sensor calibration models. Each sampling period spans 14 or 15 days, except for patient P4, whose period extended to 19 days. No diffusion tubes could be installed in the house of patient P1 due to a delay in the delivery. No outdoor data in the house of patient 2 is shown as it was lost due to a storm. Considering the measurements of the diffusion tubes as the "true value", it is evident from Fig. 9 that the SVR model predicts indoor NO$_2$ poorly, with concentrations higher than 18 µg m$^{-3}$ in all the cases. This occurs

despite achieving similar levels of uncertainty and better performance metrics as other models for the same averaging time during the testing period (see Section 3.1). RFR tends to overestimate the results, particularly for the indoor concentration measured in the house of patient P6 (average of both periods 35 µg m$^{-3}$ of $NO_2$ compared to 8 µg m$^{-3}$ measured with diffusion tubes). These discrepancies suggest that SVR and RFR overfitted the training data. The negative average values of the MLR model deployed in the house of patient P6 indicate a signal drift. Both SVR and RFR also tend to overestimate outdoor $NO_2$

concentrations, although this tendency is less pronounced compared to indoor predictions. The MLR model sometimes overestimates and sometimes underestimates the concentrations. ANN appears to be the most robust model for both indoor and outdoor sensors, even though it occasionally overestimates the actual $NO_2$ concentrations (up to 5 µg m$^{-3}$ more than the diffusion tubes).

Figure 9 (a) also shows the results of the AQSS calibrated outdoors but used indoors in the house of patient P7. When analysing

closely the outcomes, we can observe that the ML models are overestimating the results compared to the results of the diffusion tubes but for SVR and RFR less than the indoor results of the other patients as the data is in this case not overfitted. The ANN is the model that better agrees with the results, showing 2 and 3 µg m$^{-3}$ more than the $NO_2$ results of the diffusion tubes for the first and the second period, respectively. The MLR underestimates the $NO_2$ concentration the first period in the house of patient P7 and overestimates in the second period. It should be noted that the warm-up period of the $NO_2$ sensor was in this

case three days, longer than usual.

Overall, this comparison underscores the importance of not relying solely on pre-deployment performance evaluations. Reference values during deployment are crucial for verifying sensor performance. In this context, diffusion tubes have proven to be a simple and effective tool to verify calibrated sensor data.

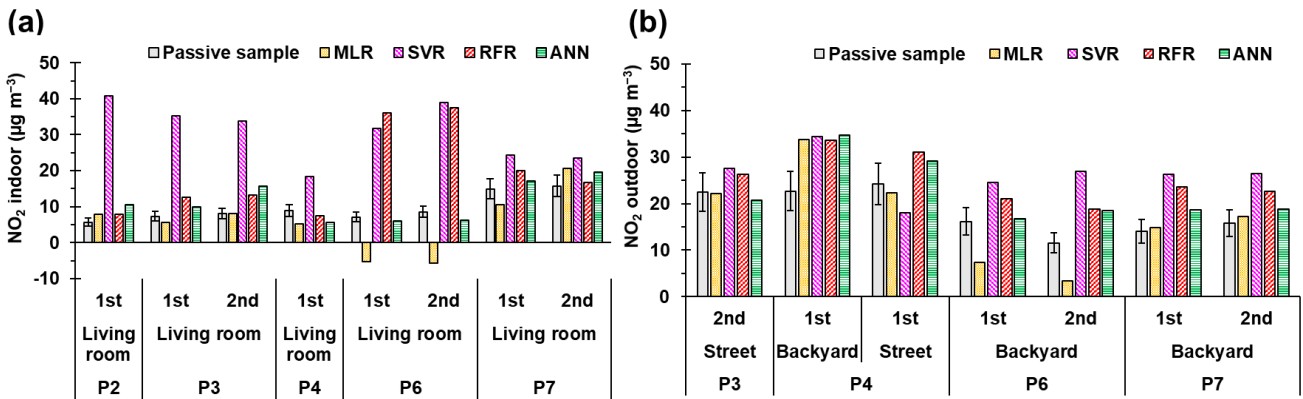

**Figure 9.** Comparison of the $NO_2$ calibration models with the concentration measured by the diffusion tubes (two-week period) for (a) indoor and (b) outdoor sensors. Models (in different colours) were trained with data averaged every 10 min. Error bars indicate the expanded uncertainty of the diffusion tubes (18.4 %).

### 3.2.2 Comparison of outdoor sensors with air quality monitoring stations

As part of the data validation process, the measurements from the outdoor AQSSs were compared with $NO_2$ and $PM_{2.5}$ data from the governmental air quality monitoring stations in the city and our measurement station at Hauptstätter Street. Figure 10 presents the results of the deployment of the AQSS placed outside the window of patient P1 and the nearest monitoring station. Additional results are provided in the Supplement (Figures S8 - S13). The calibration models for $NO_2$ sensors were trained with 10-min time aggregations.

The data of the monitoring station shown in Fig. 10 is located at Arnulf-Klett-Platz, 1.1 km from the AQSS location, near a busy road. In contrast, the outdoor AQSS was installed at the window of a second-floor apartment adjacent to a secondary road. Due to the different locations, comparisons should be approached with caution, although similar temporal patterns in the pollution concentration are expected due to the shared urban and rural background concentrations.

  Different trends in the $NO_2$ concentrations of the tested models are shown in Fig. 10 (a). Notably, the RFR model
underperforms, exhibiting excessively constant $NO_2$ levels over extended periods. This suggests that RFR is not a suitable calibration model for our study. Conversely, the SVR model fails to detect $NO_2$ concentrations below 20 µg m$^{-3}$, likely due to its limited extrapolation capability. The ANN model generally demonstrates satisfactory performance. Both the ANN and MLR models display trends that closely match the expected concentration trends. However, for other patients, MLR prediction reaches negative peaks up to $-100$ µg m$^{-3}$ (see Fig. S11). The negative peaks occurred when the T was above 25 °C. The
calibration period covered a T and a RH range of 2 - 25 °C and 40.8 - 77.4 %, respectively. However, during the measurement campaign in the house of patient P6, the $NO_2$ sensor was exposed to T up to 31 °C and RH as low as 8 %, which were far beyond the ranges covered during the calibration period. The MLR model must be used cautiously for T above 25 °C, as the influence of the T and the RH on the sensor signal is not linear (Samad et al., 2020).

  Figure 10 (b) shows that the $PM_{2.5}$ sensor equipped with a low-cost dryer and calibrated using ULR closely follows the trend
of the nearby reference station. A detailed examination reveals that the $PM_{2.5}$ readings were more accurate at the beginning of the deployment period compared to the end when the calibrated sensor reported higher concentrations than those from the reference station. Although initially unexpected, this discrepancy could be attributed to the highly localized nature of particulate matter concentrations. The placement of the AQSS in a building corner, which disrupts airflow, and its proximity to a tram line and the entrance of a hospital parking, might result in higher concentrations. If there is one field where sensors
have proven valuable, that is in identifying new pollution hotspots (deSouza et al., 2022).

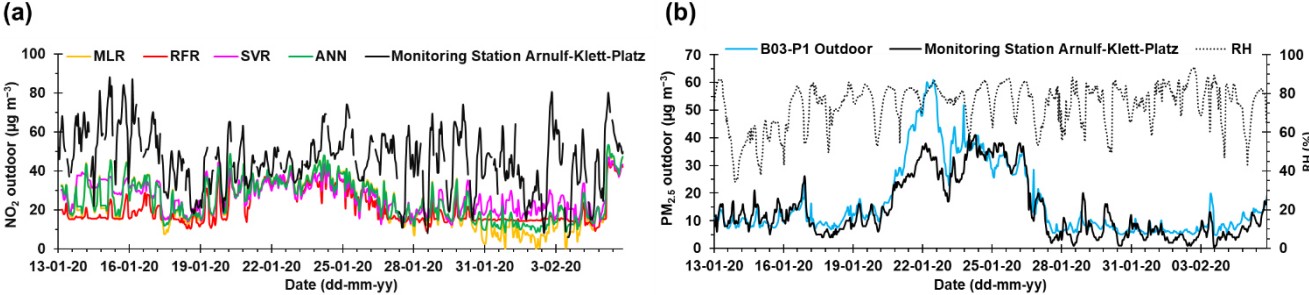

**Figure 10.** Time series of hourly outdoor (a) NO$_2$ calculated using the four tested calibration models (in different colours MLR, RFR, SVR and ANN) and (b) PM$_{2.5}$ concentrations calculated using the ULR calibration model during deployment in the house of patient P1 as well as the RH. The data of the RI is shown with a solid black line in both graphs.


A highlight from Fig. 10 (b) and Fig. S13 is that all the PM$_{2.5}$ sensors show similar trends compared to the monitoring station at Hauptstätter Street ($0.40 < R^2 < 0.93$), even the hours when the RH is higher than 70 %. The overestimation of the PM concentration by the sensors at high RH due to the hygroscopic growth of particles is avoided thanks to the thermal dryer.

### 3.2.3 Metadata for qualitative sensor data validation

In this section, we present an example of how the use of metadata, specifically the activity and window status logs, can be used as a complementary tool to validate and understand sensor data in places without RI. Figure 11 shows the indoor NO$_2$ and PM$_{2.5}$ concentrations during the second week of deployment in the house of patient P2. Additionally, the different activities on an hourly basis and the status of the windows in the living room where the AQSS was located are shown. The NO$_2$ sensor readings have been corrected using the ANN model based on 10-min aggregation time.

As illustrated in Fig. 11, pollution peaks can be correlated with specific activities at home. The information collected in the logbook is invaluable for interpreting sensor data. It allows for the detection of anomalies and helps in understanding the source of pollution peaks. For instance, in Fig. 11 (c), there is a noticeable decrease in PM$_{2.5}$ concentration during sleeping hours and an increase during activities like exercising (on 24 January 2020) and cooking (on 24 and 27 January 2020). For NO$_2$, the activity log is especially useful when considering window status, as NO$_2$ typically originates from outdoor sources in houses

with electric stoves. This is evident in Fig. 11 (b), where some peaks occur when the window is open or tilted. A deeper analysis of the information acquired in the log books and the relationship with the indoor air quality in the houses of the patients can be read in Chacón-Mateos et al. (2024). Other studies, such as that by Novak et al., have proposed methodological frameworks that more systematically integrate metadata from activity logs with air quality sensor data (Novak et al., 2024; Novak et al., 2023a; Novak et al., 2023b).


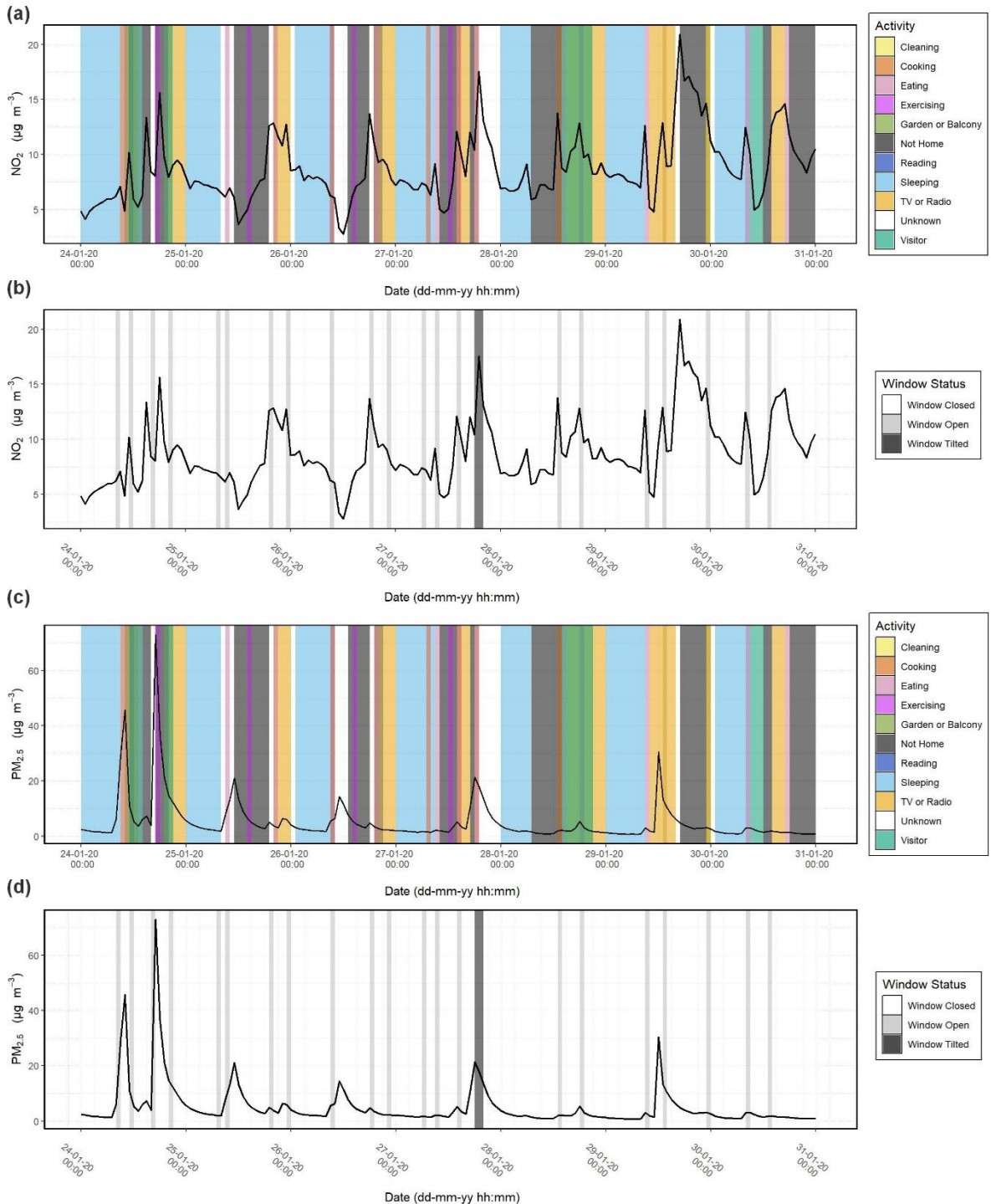

**Figure 11.** Hourly times series of (a) indoor $NO_2$ concentration and activities, (b) indoor $NO_2$ concentration and window status, (c) indoor $PM_{2.5}$ concentration and activities and (d) indoor $PM_{2.5}$ concentration and window status during one week in the house of patient P2.

## 4 Discussion

### 4.1 Evaluation of the NO₂ sensors

#### 4.1.1 Indoor NO₂ sensors

The results of this study indicate that using indoor co-location and artificially generated $NO_2$ to correct the signal of electrochemical sensors for $NO_2$ may not be effective for the tested models as it can cause model overfitting. Calibrating indoor sensors presents particular challenges due to two main factors: the low concentrations and the need to train models across different spans of concentrations, T, and RH. Although the testing data showed nearly perfect results after training the sensor data with artificially generated $NO_2$ and controlled changes of T and RH, applying the model to the sensor data deployed in the patients' homes yielded significantly different outcomes. Some models like SVR and RFR struggled to accurately predict the $NO_2$ concentrations in the new indoor environment as they overfitted the training data.

We conducted an experiment to test whether an AQSS for indoor use could be calibrated in outdoor co-location to better learn real $NO_2$ concentration and meteorology patterns. Although the calibration models tended to overestimate concentrations compared to the diffusion tubes, the SVR and RFR models did not exhibit overfitting, unlike what was observed with the indoor calibration. Note that in Stuttgart outdoor $NO_2$ concentrations are generally higher than indoor concentrations, and the models are not adept at extrapolating to lower concentrations. That represents a challenge for using an outdoor co-location to calibrate a $NO_2$ sensor for indoor measurements.

Other solutions for indoor sensor calibration could be a hybrid calibration like the Enhanced Ambient Sensing Environment (EASE), which combines the advantages of laboratory calibration with the increased accuracy of field calibration (Russell et al., 2022). To date, this approach has only been tested with multilinear regression models and in outdoor environments. Further research is needed to determine whether it is suitable for indoor environments and the training of machine learning algorithms.

Another possible solution is the calibration of the sensors in occupied homes (Suriano and Penza, 2022) or exposing the sensors to cooking events (Tryner et al., 2021). However, these studies did not deal with the re-colocation of the monitors after the calibration in a new environment. Therefore, further research is needed to expand our knowledge of calibration transfer in indoor environments for electrochemical sensors.

#### 4.1.2 Outdoor NO₂ sensors

The calibration of $NO_2$ sensors through a co-location with RI outdoors is at this moment a common procedure (Karagulian et al., 2019). Many studies have tested different regression and ML models (Spinelle et al., 2015; Cordero et al., 2018; Zimmerman et al., 2018; Malings et al., 2019). Our results on the performance evaluation for the outdoor $NO_2$ sensors are similar to the outcomes from Bigi et al. (2018) obtained using 10-minute averages for MLR, SVR and RFR and to those from Apostolopoulos et al. (2023) for the ANN model (note that their results are based on hourly values).

One limitation of our study was the lack of ozone data. It has been demonstrated that the sensor B43F has cross-sensitivity to ozone despite having an ozone filter and that the influence of ozone increases as the filter saturates (Li et al., 2021). Knowing

this, we experimented with adding ozone data from the air quality monitoring station located at Marienplatz (Stuttgart) to train the calibration models of B03-P1 and B05-P7, representing the cold and the warm months, respectively. The results of the error metrics are shown in Fig. S14. Even though the results of the $R^2$ and RMSE seem to improve in most of the cases adding ozone data, the results of the MAE show the opposite trend in the cold months (P1). Moreover, the difference in the RMSE results during the warm month (P7) is minimal for most of the models except for MLR. Therefore, we did not further investigate the addition of ozone data for the rest of the data. Furthermore, ozone concentrations are higher in summer months and our measurement campaign ran from December to May, i.e. mainly in winter months when ozone concentrations are lower. In addition, studies have shown that the performance degradation of the ozone filter starts 200 days after sensor unpacking (Li et al., 2021), which is approximately the number of days that our campaign lasted. Nevertheless, for future studies, we recommend adding an ozone sensor so the cross-sensitivity can be corrected for all seasons.

Moreover, we would like to highlight that incorporating data from a neighbouring station as an input feature for training the calibration models was identified as a "questionable parameter" by Hagler et al. (2018), as it may compromise data integrity, blurring the line between an actual measurement and a model prediction (Hayward et al., 2024).

### 4.1.3 Evaluating averaging times

The choice of the temporal resolution significantly affects the quality of training data for $NO_2$ sensor calibration models. Even if the number of data points using a temporal resolution of 1 min was between 28,000 and 5,100 for indoor calibrations and between 14,400 and 5,500 for outdoor calibrations (see Figure S5), these high-resolution data contained more noise, which impacted negatively the training quality. Conversely, using coarser resolutions (e.g., hourly averages) may excessively reduce the number of training data points available and the concentration range covered in the calibration. Our study found that using a 10-min averaging period over a two-week calibration phase (comprising between 400 and 2,500 data points) resulted in a lower MAE for $NO_2$ sensors. However, the difference compared to a 15-min averaging period was small across most metrics, including target diagrams, REUs and the comparison with the $NO_2$ concentrations measured by the diffusion tubes. Although some researchers have employed hourly averages (Cai et al., 2009; Wei et al., 2020; Goulier et al., 2020), others have also identified a 10-minute average as the optimal (Paas et al., 2017; Bigi et al., 2018). In contrast, Sahu et al. (2021) in their analysis of the effect of temporal data averaging, found that data averaged every 5-min provided better results. All in all, the selection of an appropriate averaging time depends largely on the quantity of available training data and must be carefully selected.

### 4.1.4 Evaluating calibration models for $NO_2$ sensors

The results of our study show that ANN is the most robust model for transferring the calibration parameters of the sensors to be used in another place, either indoors or outdoors, using the proposed calibration methodology. Even though RFR and SVR show better results for the metrics RMSE, MAE and Pearson correlation coefficient and similar REU and target diagram results to ANN and MLR during the calibration phase prior to deployment, the comparison with the diffusion tubes during the

measurement campaign in the houses of the patients showed that SVR and RFR overestimated in most cases the $NO_2$ concentrations. The MLR showed the worst performance among the tested models.

### 4.2 Evaluation of the PM$_{2.5}$ sensors

#### 4.2.1 Indoor PM$_{2.5}$ sensors

Validating PM$_{2.5}$ sensor measurements indoors presents significant challenges. While activity logs provide invaluable information regarding events that might cause elevated PM$_{2.5}$ concentrations, it remains unclear without having a RI in the houses, whether sensors can accurately quantify these peaks. This uncertainty may be particularly problematic in short-term exposure studies, where precise measurement of peaks is critical. However, in long-term studies, short-duration peaks contribute less to the overall concentration average, thus presenting a lesser concern.

The results of this study suggest that using test aerosols like liquid paraffin in a particle chamber may not be an optimal technique for PM sensor calibration. This is likely due to discrepancies between the density assumed in the sensor's internal algorithm and the actual density of the generated particles.

Our study also explored the use of an outdoor calibrated AQSS intended for indoor deployment. However, due to the lack of an RI indoors during deployment, we cannot conclusively determine if this method outperformed indoor calibrations carried out in the laboratory. Previous research, such as the study by Koehler et al. (2023), suggests that calibrations using ambient outdoor air data can enhance the data quality of indoor sensors compared to using manufacturer-provided calibrations. Nonetheless, the composition and concentration ranges of PM indoors can significantly differ from that of outdoor air, which may affect the correct performance of the sensor calibration. Further research is necessary to evaluate various calibration methods for indoor sensors and to understand how different PM compositions influence sensor performance.

#### 4.2.2 Outdoor PM$_{2.5}$ sensors

One of the biggest concerns about PM sensor measurements outdoors is the effect of hygroscopic growth or fog. The use of either physical air preconditioning or data post-processing considering the RH is a must in regions where high relative humidity and hygroscopic aerosols are expected, as it is the case of Stuttgart. For this project, a low-cost dryer unit was designed to avoid the overestimation of PM$_{2.5}$ concentrations.

The results of the comparison of sensor data with data from local monitoring stations in Stuttgart in the vicinity of the houses of the patients showed that the PM$_{2.5}$ sensors showed a similar trend even when the RH was higher than 70 %. Given the fact that a simple linear regression applied to the outdoor PM$_{2.5}$ sensors with a dryer shows plausible results when compared to the nearest measurement stations, this method can be used to simplify the models for PM calibration. However, it is important to control the drying temperature as temperatures higher than 40 °C could evaporate semi-volatile organic compounds and trigger the underestimation of the PM mass concentration (Chacón-Mateos et al., 2022).

### 4.3 Do the sensors fulfil the Data Quality Objectives?

Previous studies have indicated that while commercially available AQSSs often meet the criteria for indicative measurements of $PM_{2.5}$, $NO_2$ sensors frequently struggle to fulfil the DQO (Castell et al., 2017). This challenge prompted the design and evaluation of our AQSSs. However, the rapid advancement in sensor technology outpaces scientific literature, making it difficult to keep up with the latest developments.

Regarding $NO_2$ sensor units, many researchers have applied calibration models that account for parameters such as RH, T, and
715 ozone data. These models have demonstrated that the DQO for indicative measurements can be achieved for $NO_2$ concentrations above 20 ppb (Spinelle et al., 2015; Bigi et al., 2018; D'Elia et al., 2024). Our findings align with these results, showing that outdoor $NO_2$ sensors meet the DQOs of both EU Directive 2008/50/EC and 2024/2881 for indicative measurements between 10 and 25 ppb, depending on the specific sensor unit and the averaging time used. Sensors calibrated in indoor conditions performed even better, achieving the DQOs at even lower concentrations. However, we have also argued
that the use of a GPT system to generate controlled $NO_2$ concentrations may not be appropriate for training ML models intended for deployment in indoor environments.

It is evident that even after calibration, the "hardware" of electrochemical sensors has not reached enough maturity yet for applications requiring low measurement uncertainty, especially for low concentrations, making the measurement very dependent on the "software" used to correct the data (regression models, ML, etc.). Recent advancements in sensor units
include onboard temperature monitoring near the electrical cell, which appears highly promising to improve the accuracy of the calibration models.

Our research also highlights the impact of the averaging time on the REUs of calibrated sensors. Generally, coarser averaging times improve the likelihood of meeting the DQO at lower concentrations, though this often reduces the concentration range covered during calibration. Moreover, ML models may not predict accurately outside the concentration range for which they
were trained.

For $PM_{2.5}$ measurements, both DQOs are in the new EU Directive 2024/2881 stricter, from 50 to 35 % for indicative measurements and from 100 to 85 % for objective estimation. Considering that, the DQO for indicative measurements after an indoor sensor calibration is typically achieved at concentrations above 23 and 35 $\mu g\ m^{-3}$ for the Directives 2008/50/EC and 2024/2881, respectively. After field calibration of the outdoor units, the DQO for indicative measurements is achieved at
735 concentrations higher than 16 $\mu g\ m^{-3}$ under EU Directive 2008/50/EC. However, four out of nine sensors fail to meet the DQO criteria of EU Directive 2024/2881. Moreover, a significant unit-to-unit variability exists. This variability has been noted in previous studies, such as those on the SDS011 sensor (Liu et al., 2019).

In summary, while the tested sensor units generally fulfil the DQOs for higher concentrations, the higher REU of the sensors at lower concentrations may hinders their application in epidemiological studies. Despite limitations at low pollutant levels,
calibrated AQSSs are a promising tool to increase the ubiquity of epidemiological studies for low- and middle-income countries or regions where higher air pollutant concentrations are expected, where more epidemiological studies are needed (Amegah,

2018). Nevertheless, it is important to acknowledge that even RI are not free from uncertainties (Diez et al., 2024). Regular quality control is essential for all air quality monitoring devices, whether they are gold standard, reference-equivalent, or sensor-based.

**4.4 The real cost of "low-cost" sensors**

In this study, we designed two AQSSs costing approximately 400 euros for indoor and 500 euros for outdoor measurements, excluding labour costs. Despite the relatively low acquisition cost compared to a RI, the implementation and maintenance of the AQSSs are not necessarily low-cost. Moreover, the use of AQSSs in health studies requires the acquisition of RI for their calibration, as well as additional time for co-location, which must be accounted during the planning phase.

Note that the term "low-cost" varies significantly by region, and we have intentionally avoided its use in this manuscript. Even though we acknowledge that the term "low-cost" or the abbreviation "LCS" has helped to differentiate them from traditional air monitors and form a recognizable community, we recommend that future publications also refrain from using "low-cost" or "LCS" and instead use "air quality sensors" or "AQS".

**5 Conclusion**

In this study, we evaluated the performance of the OPC-R1 and the B43F sensor models for measuring $PM_{2.5}$ and $NO_2$, respectively, for their use in health studies across both indoor and outdoor microenvironments. For that purpose, we used REUs, target diagrams and common error metrics. A central research question concerned whether calibrated sensors could meet the DQOs defined in the EU Directive 2008/50/EC and in the recently published EU Directive 2024/2881, and if so, at which concentration levels.

The co-location phase was conducted two weeks before the deployment, where the data from RI were used to calibrate the $PM_{2.5}$ sensors with ULR and test regression (MLR) and ML models (RFR, SVR and ANN) to calibrate the $NO_2$ sensors. The results show that the REUs depend on the temporal average (*i.e.* the number of data points) used during the training. Generally, coarser averaging times (10 and 15 min) improved the likelihood of meeting the DQO at lower concentrations while high-resolutions (1 and 5 min) led to higher REUs due to the impact of the sensor noise in the training data.

The validation of the sensor data during deployment in the houses of the patients was performed using $NO_2$ diffusion tubes, patient logbooks with activity information and window status as well as data from the monitoring stations in Stuttgart. Even though ML seems a promising tool in the field of AQS, the training data acquired by exposing the sensor and the co-located RI to artificially generated $NO_2$ for indoor calibration did not yield realistic results (compared to the $NO_2$ measurements of the diffusion tubes) for some of the ML models (RFR and SVR). Furthermore, performance evaluation revealed that calibrating PM sensors using liquid paraffin as a test aerosol is problematic, owing to mismatches between the assumed particle density in the sensor's internal algorithm and the actual density of the generated aerosol.

Our results highlight that the environmental conditions (e.g. temperature and relative humidity ranges) and concentration levels present in the training phase are critical for ensuring reliable data when sensors are relocated. The choice of temporal averaging used to train the models directly affects the range of concentrations, temperatures and RH covered and, consequently, it has a direct impact on the performance of the calibration model. Moreover, the integration of metadata, such as activity logs, window status, data from official monitoring stations and diffusive samples, was proved a good tool for validating and interpreting sensor data.

There remains a need for more comprehensive sensor evaluations that extend beyond basic statistical metrics such as $R^2$ and MAE. Tools like REUs and target diagrams add significant value by enhancing trust and transparency in sensor data. Future work should also prioritise assessing the transferability of calibration models, particularly those developed in indoor co-location settings, to enable the integration of reliable and traceable air quality sensor data in future health studies.

## Data availability

The data of this study are available from the authors upon request.

## Author contributions

Conceptualization, M.C.-M., U.V. and B.L.; data curation, M.C.-M. and H.G.S.; formal analysis, M.C.-M.; funding acquisition, U.V. and B.L.; investigation, M.C.-M., B.L. and H.G.S.; methodology, M.C.-M., B.L. and U.V.; project administration, U.V.; resources, B.L. and U.V.; software, B.L. and H.G.S.; supervision, B.L. and U.V.; validation, M.C.-M.; visualization, M.C.-M. and H.G.S.; writing—original draft, M.C.-M.; writing—review and editing, M.C.-M., B.L. and U.V. All authors have read and agreed to the published version of the manuscript.

## Competing interests

The authors declare that they have no conflict of interest.

## Acknowledgements

Grecia Carolina Solis Castillo and Ioannis Chourdakis are highly acknowledged for their valuable contribution during the project. Frank Heimann is acknowledged for helping to recruit the participants. Erika Remy contributed to create Figure 11. Finally, we thank all volunteers who participated in the study.

**Financial support**

This work was supported by the Ministry for Social Affairs and Integration Baden-Württemberg (grant no. AZ 53-5425.1/5).

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
