# Peer review of "Figure S1. Map of Stuttgart showing the locations of the participants' homes where sensors were deployed (yellow diamonds), the governmental outdoor air quality monitoring stations (blue circles) and the monitoring stations that belong to the University of Stuttgart (green circles)."

_EGUsphere, 2025_

## Referee Comment (RC2)

**General comments**

This manuscript addresses a highly relevant and timely topic: the evaluation of low-cost sensor performance in real-world exposure settings, with a focus on their potential application in epidemiological studies. The proposed approach is valuable, and the work presents several strengths, including the use of a modular sensor platform, the incorporation of multiple performance metrics, and the attempt to link results to regulatory benchmarks. In this regard, the study has the potential to become a significant contribution to the literature on sensors and personal exposure assessment.

However, the manuscript still requires certain adjustments to reach its full impact. Some elements of the study design are not fully integrated into the discussion of the results, and certain sections would benefit from greater methodological clarity. Overall, the manuscript feels somewhat disorganized, with sections that could be restructured to enhance internal coherence, and the use of technical language could be refined to improve precision. Finally, some conceptual discussions are potentially valuable, but appear somewhat disconnected and would benefit from being more clearly linked to the study's main objectives and findings.

In summary, this is a manuscript with an original approach and strong potential, which could be significantly strengthened through a comprehensive revision aimed at improving its structure, clarity, and argumentative coherence.

**Comments by section**

**Title**

The current title partially reflects the rich content of the study. In particular, it would be important to highlight that the study does not exclusively address performance, but also includes the calibration process of the sensors, and that its main value lies within the framework of environmental epidemiology A suggestion could be "Calibration and performance evaluation of low-cost PM2.5 and NO2 air quality sensors for environmental epidemiology".

**Abstract**

The abstract effectively summarizes the main elements of the manuscript, but some rewording could improve its clarity and alignment with the content of the paper. For instance, the second sentence ("This study aimed to transcend the limitations...") may come across as somewhat broad and ambitious, and could benefit from a more specific description of the limitations being addressed. The final sentence of the abstract reads more as an aspirational statement than a conclusion directly supported by the findings. It may be more appropriate to include this type of reflection in the discussion section.

**Introduction**

The introduction correctly outlines the distinction between direct and indirect approaches to assessing exposure to air pollutants. However, when discussing the direct approach, the text focuses primarily on mobile personal measurements, which does not accurately represent the methodology used in this study.

Given that a substantial portion of the study focuses on evaluating sensor performance in indoor environments, it would be useful to include in the introduction a more explicit review of previous work in this area. For instance, studies that have assessed the performance of sensors

in domestic settings or proposed calibration/validation strategies specifically tailored to indoor conditions. Not including this indoor background limits the reader's ability to grasp from the outset how underdeveloped the literature is in this domain. It also reduces the opportunity to position this study as an important contribution to a still-emerging field, and to contrast the findings with relevant previous work.

One of the most valuable aspects of the manuscript is the use of regulatory metrics, such as the calculation of relative expanded uncertainty (REU) as defined in European directives, to assess sensor performance. However, these directives, and the related technical specifications, were not designed for personal monitoring. This represents an opportunity for the manuscript to elaborate more thoroughly on an important gap in the literature: the relevance, limitations, and potential of using regulatory standards as quality criteria in epidemiological studies. Addressing this point would strengthen the conceptual justification of the chosen approach and connect it to recent debates on the adaptation of regulatory frameworks to emerging technologies.

**Methodology**

Section 2.1 includes valuable elements of the study design, such as pulmonary function tests, health surveys. However, these elements are not developed or connected with the results presented. It would be useful to clarify whether these data fall outside the scope of this manuscript or if they will be analyzed in future work. Otherwise, the reader may perceive a disconnect between the protocol and the actual focus of the study.

The description of sensor deployment would benefit from more detailed information on installation conditions, such as the height above ground, orientation and distance from potential sources (e.g., windows, heaters, traffic), and the ventilation characteristics (passive or active?) of the protective enclosure.

The title "Study design and quality assurance" could be revised, as subsections 2.3.1 and 2.3.2 focus on quality control rather than experimental design.

The title of subsection 2.3.1, "Sensor calibration before deployment," also seems inappropriate (same with the title of Table 1), since the content primarily addresses the co-location experiments necessary for calibration but does not include the full calibration process, which is described in other sections (e.g., 2.3 Sensor correction models). Reorganizing these subsections could improve clarity and support reader comprehension.

Some passages in subsection 2.3.1 could be edited for clarity, as they currently lack precision. For example, statements like "The experiments in the particle chamber took about an hour and were repeated at least twice" or "the sensors were co-located in the laboratory for some days" are somewhat vague.

Since the selected co-location site (Hauptstätter Straße) is a known urban hotspot with elevated pollution levels, a brief methodological justification for this choice would be useful.

Throughout the manuscript, the term "reference instruments" (RIs) is used generically, but under European legislation there are two levels of "fixed measurements": reference instruments and equivalent to reference. It would be helpful to specify which category applies to each instrument used in this study.

Further clarification is also recommended in section 2.3.2: Which sensors were validated against which official stations? How far apart were they? How was agreement with the passive diffusion tubes quantified? What interpretive criteria were applied for indoor PM2.5 records? Table 2 indicates that the validation with diffusion tubes was quantitative, how was the quality of those measurements ensured?

The title of section "2.3 Sensor correction models" does not accurately reflect the content, as the methods described (MLR, SVR, RFR, ANN) are in fact calibration procedures. The manuscript uses the terms "correction" and "calibration" somewhat interchangeably. Since the study has a strong methodological focus on performance evaluation and the application of statistical calibration models, more precise terminology is recommended. I suggest referring to the International Vocabulary of Metrology (https://www.bipm.org/documents/20126/2071204/JCGM_200_2012.pdf). It would also be helpful to include a summary table that, for each pollutant and environment combination, specifies: the model used, the explanatory variables, the reference instrument type, and details on training, testing, and validation. What temporal resolution was used to train the models?

Section 2.4 indicates that the models were trained using 15-minute aggregated data, while section 2.5 reports performance evaluated at 1, 5, and 10 minutes. It is important to clarify whether models trained at 15-minute resolution were directly applied to data at other resolutions, and what the limitations of this approach are.

In section 2.5, where the use of target diagrams is described, a more stringent performance criterion is defined using a threshold of 0.5. However, this threshold appears arbitrary. It would be helpful to justify its selection, or to clarify that it is an exploratory criterion adopted for this study.

This section combines references to technical standards CEN/TS 17660-1 (2021) and 17660-2 (2025), which are specific to sensors, with DQOs from EU directives 2008/50/EC and 2024/2881, which were not designed for low-cost devices or indoor measurements. This could mislead readers (especially those unfamiliar with EU regulation) into thinking that sensors are expected to comply with the stricter DQOs from the 2024/2881 directive, which is not the case in current practice or within the scope of existing CEN/TS guidance. It is also important to note that both the CEN/TS standards and the cited directives are aimed at outdoor ambient air quality and set minimum temporal resolutions of one hour or longer (depending on the pollutant). Therefore, their direct application to shorter averaging intervals, like those used in this study, is not formally covered. It is suggested to explicitly state whether the inclusion of new DQOs is intended as an exploratory or forward-looking exercise, and to clarify that the current regulatory framework remains anchored in the 2008/50/EC directive and its associated DQOs. Simplifying the language in this section could also help improve accessibility for non-specialist readers.

The statement in line 341 regarding the need for "enough data points" to train the models is relevant but not further developed in the text. It would be valuable to include a more explicit discussion of the actual number of data points used in this study for calibrating each model (especially the NO2 models based on machine learning), and whether that amount is adequate given the complexity of the algorithms. Additionally, the expression "low uncertainties" may be ambiguous to some readers, as it refers here to the uncertainty associated with the calibration model (i.e., how well the functional relationship between raw and calibrated data reflects the actual relationship), and not to total measurement uncertainty (of which calibration

uncertainty is only one component). Clarifying this distinction would help avoid potential misinterpretation.

**Results**

In section 3.1.1, while PM2.5 was modeled using a linear approach with modest data volumes (200-600 data points per sensor), the NO2 models rely on machine learning algorithms (MLR, SVR, RFR, ANN), which typically require significantly larger datasets to avoid overfitting and ensure model stability. Since no table equivalent to Table S5 is provided for NO2, and the data volumes used are not explicitly reported, it is difficult to assess the validity and robustness of the applied models. See also the related comments on section 2.5.

In section 3.1.2, all NO2 calibration models fall on the left-hand side of the target diagram. It would be relevant to consider whether this reflects a systematic underestimation of the actual variability by the calibrated sensors, or a limitation in the dynamic response range of the devices.

The inclusion of multiple additional performance metrics in this section (MAE, MEF, slope/intercept, r) does not appear to be clearly justified, especially given that robust tools such as REU and target diagrams (which integrate error and bias) are already applied. Without a clear explanation of their added value (particularly from the perspective of potential users such as researchers, agencies, or communities) these metrics may come across as a statistical exercise without a defined purpose. Moreover, their inclusion could obscure the central message of the manuscript. It is recommended to focus this section on the key metrics relevant for practical decision-making and consider moving the rest to the Supplement.

In section 3.2.1, passive samplers are used to validate sensor performance; however, no information is provided about the quality/uncertainty of the data they produce.

Section 3.2.3 presents the use of activity logs as a strategy to contextualize sensor data and explore individual exposure patterns. It is suggested to frame this as an exploratory or complementary tool, and to consider whether previously cited studies offer methodological frameworks that could support a more systematic integration of metadata in future research on personal exposure.

**Discussion**

The final paragraph of section 4.1.1 introduces the "EASE" methodology, but it is unclear what exactly the authors are referring to, which hinders its interpretation. Additionally, this paragraph includes some generalizations that do not appear to be directly supported by the study's results. It is recommended to revise the paragraph to more accurately reflect the exploratory nature of this reflection.

Section 4.1.2 would benefit from being restructured to clearly distinguish between: (i) the review of previous studies, (ii) the authors' own findings, and (iii) the methodological decisions taken. It would also be useful to discuss whether the use of a linear model to incorporate ozone was the most appropriate choice, considering that sensor cross-sensitivity to O3 is not necessarily linear or constant.

Section 4.1.3 addresses the effect of averaging time on NO2 sensor calibration, but the empirical evidence is limited due to the short duration of the calibration phase (two weeks). It is recommended to report how many data points were used for each averaging interval (1, 5,

10, 15 min) and to discuss how variability in sample size affects the comparison. The statement "Our study demonstrated…" could be softened, as the available data do not support a generalizable conclusion. Moreover, the reduction in MAE is not contextualized in relation to other performance metrics, nor is its practical relevance evaluated, which makes it difficult to interpret the actual benefit of using a 10-minute average.

One of the recommendations in section 4.3 states: "we recommend adjusting the averaging time based on the available data...". This may be confusing or lead to methodologically questionable interpretations. The choice of temporal resolution should primarily be guided by the study's objectives (e.g., capturing acute exposure events), not just by the amount of data available. It is suggested to rephrase this recommendation or to justify its applicability in more detail.

The closing paragraph of this section suggests that, due to the poor performance of sensors at low concentrations, they may be more suitable for environments with high pollution levels. However, this statement is not supported by empirical evidence in the present study. It is recommended to revise this conclusion to more directly acknowledge the limitation observed, rather than inferring a potential use case that was not investigated here.

The reflection in section 4.4 on the meaning of the term "low-cost" is valid, but feels disconnected from the rest of the manuscript. If this content is to be retained, it is suggested to move it to the conclusions and, most importantly, to explicitly connect it to the study's findings. For instance, the authors could discuss whether the calibration, validation, and performance requirements observed in this study still justify labeling these sensors as "low-cost" in real-world applications. Otherwise, the section may come across as tangential or anecdotal.

**Conclusions**

The conclusions section brings together various elements of the study, but does not clearly articulate a central message regarding the value, applicability, and limitations of the evaluated sensors for epidemiological research. It is recommended to revise this section to: (i) avoid generalized claims that are not supported by the evidence presented, (ii) strengthen the critical analysis of the results (e.g., the limited transferability of laboratory calibrations, or the actual utility of machine learning models with small datasets), and (iii) better guide the reader regarding practical implications and future directions. Reformulating the conclusions based on the study's initial objectives would help close the manuscript with greater clarity and strength.

**Specific comments**

Line 29: Consider adding "fine fraction" when referring to PM2.5.

Lines 45-46: Suggest adding a reference to support the statement.

Line 62: It would be useful to acknowledge that, although the high temporal resolution offered by sensors is valuable for capturing dynamic exposures, it also increases instrumental noise, which directly impacts measurement uncertainty, particularly important in epidemiological studies.

Line 82: It may be appropriate to introduce that this is a study attempting to apply a direct measurement approach.

Lines 88-89: Consider rephrasing for improved clarity.

Lines 95-96 and 103-104 could be rewritten for a more natural and concise expression.

In section 2.2 (and in other parts of the manuscript), the terms "air quality monitor", "sensor system" and "AQSS" are used interchangeably without clarifying whether they are synonymous.

Check the numbering of the "Study design and quality assurance" section (2.3?).

In Table 1, clarify what is meant by "low concentrations".

Lines 161-162: Clarify what the authors are referring to.

Lines 249-250: The wording could be improved for clarity and precision.

Line 253: This statement could be reconsidered or qualified: "In theory, the higher the number of data points, the better the model performs", as model performance does not necessarily with more data points.

Lines 270-271: "Higher performance" could be replaced with "better performance" or a more neutral formulation. Also, "in general" suggests there may be exceptions, but that's not applicable here, where the interpretation of target diagrams is systematic.

Line 287: "Resolution" appears twice.

Line 288: The phrase "especially in mobile measurements" seems out of place in the context of this study, since no mobile measurements were performed. In the following sentence, there may be a typo ("long" instead of "longer"?).

Consider including a brief explanation of the nomenclature used for sensor IDs.

Figures are in general a bit difficult to interpret, are different models, averaging intervals, or measurement phases being shown? It is suggested to include a complete legend in each figure or, alternatively, clearly describe in the text what each marker represents.

Figure 8 does not show results for participants P1 and P3, and the text in section 3.2.1 does not explain their exclusion. The sampling period (14 or 15 days, as mentioned in the text) is also not indicated. Recommend expanding or clarifying.

The term "passive samples" is used throughout the text. A less ambiguous expression could be "passive sampler measurements" or "passive sampling data".

Line 455: It seems to refer to the data collected by the monitoring station, not the station itself. Check for clarity.

Line 456: It is recommended to specify more clearly that the sensor was located outdoors, outside the second-floor window, to avoid ambiguity.

Lines 464-465: It is stated that this is a "clear example" of the effect of temperature and relative humidity on electrochemical sensors. However, the presented data do not allow us to determine whether the observed error is due to sensor limitations or extrapolation of the model beyond the training range.

In Figure 9b (PM2.5), relative humidity is included as a contextual variable, which may be misleading since, according to the methodology, RH was not used as a variable in the PM calibration model. Given that RH was used for NO2 models (Figure 9a), this graphic choice could be misleading and would benefit from clarification.

Line 470: The statement that machine learning models "consistently overestimated" concentrations compared to passive tubes may be overstated, as only one biweekly average value per participant is available, and the comparison is limited to a few cases (Figure 8).

The use of the term "addition of ozone" in lines 531 and 535 could be misleading, as in the context of atmospheric chemistry or sensor testing it is often interpreted as the physical addition of ozone gas to an experimental mixture.

---

## Author Comment (AC1)

**Response to interactive comments from Referee #1**

Thank you for the time you put into reviewing our manuscript and the helpful feedback. Please see our following responses and proposed changes to the original manuscript, which we believe, help to improve this paper and increase its impact. Below the comments from Referee #1 are given in black. Our responses to the comments are shown in blue. Text added or changed in the manuscript is marked in italics.

First of all I would like to congratulate the author for this very interesting and very complete work on the real use of sensor system in an epidemiologic study. The use of the uncertainty as a marker for reliability is a very interesting choice. However, I would like to emphasize that the European Directive DQO uses a specific value usually taken as the limit value set by the Directive itself. Nevertheless, the work presented in this paper has been carried out with great care resulting in a very nice publication.

We sincerely thank the reviewer for the kind words and thoughtful feedback on our work. We greatly appreciate the recognition of our efforts in applying sensor systems within an epidemiologic framework, as well as the interest in our approach to using the relative expanded uncertainty to evaluate of the sensor performance.

Regarding the valuable observation on the European Directive DQOs (Data Quality Objectives), we acknowledge the importance of aligning with the directive's specifications, particularly in reference to the limit values. However, the DQOs of EU Directive 2008/50/EC and 2024/2881 as well as the mentioned CEN/TS 17660-1 (2021) and 17660-2 (2024) set minimum temporal resolutions of 24-hour, 8-hour and hourly (depending on the pollutant) for short-term mean concentrations. In our study, due to the limited time we had for the indoor and outdoor calibrations we tested temporal resolutions of 1, 5, 10, and 15 min for $NO_2$ sensors and 1 and 30 min for $PM_{2.5}$ sensors, for indoor and outdoor respectively. This differ from the temporal resolution stated in the EU Directive 2008/50/EC and 2024/2881 to calculate the relative expanded uncertainty.

Moreover, in the CEN/TS 17660-1 (2021) and 17660-2 (2024) were our study takes the inspiration from, strict requirements are defined for the tests: sensors must be co-located at least two different type of stations, two seasons per type of station and 40 days per campaign. As the full project had a length of 1.5 years including design of two sensor systems, calibration, deployment and evaluation, it was not possible to co-locate the sensors with reference equivalent instruments for more than 2 weeks before each deployment. The attempt of evaluating the REU in the limit values (both the current or the future ones) could lead to a misinterpretation of the sensor classification.

We have modified the following text in the manuscript to acknowledge this limitation:

*"Note that the short-term DQO EU Directives were conceived for hourly and daily averages for $NO_2$ and $PM_{2.5}$, respectively. For epidemiological studies, however, especially those using portable monitors, 24 h average or even 1 h average may be insufficient, as detecting short-term pollution peaks requires higher temporal resolutions. Moreover, longer co-location periods are not always possible during the exposure assessment campaigns and consequently, the use of a 1-hour average can decrease considerably the available data to train the calibration models and reduce the range of T and RH, as well as the pollution concentration range used. Therefore, in this work, we present the REUs of the $NO_2$ models for different averaging times, that is, 1, 5, 10, and 15 min and thus, an evaluation of the REUs at the limit value is not applicable. Similarly, co-location measurements of indoor $PM_{2.5}$ sensors in a particle chamber with high particle concentrations lasted an average of 2 to 3 hours. Therefore, the uncertainties were calculated for a 1-min average. For outdoor $PM_{2.5}$ sensors where more data points are available, a 30-min average was used so that neither REU for $PM_{2.5}$ measurements for indoor or outdoor are applicable in the region of the limit values."*

You can find below some comments I had on some specific point:

- Table 2: in the comparison with outdoor AQMS, how did you define the 6km limit?

It is not a real limit, it represents the air distance of the house located the furthest from the monitoring station, that is, 6 km away. We have clarified this in the text as follows:

*"Additionally, the data of the four outdoor air quality monitoring stations available in Stuttgart as well as the data of the monitoring station of the University of Stuttgart in Hauptstätter Street was also collected to qualitatively compare their $NO_2$ and the $PM_{2.5}$ trends with the data of the outdoor AQSSs during deployment in the houses of the patients. The air distances between the closest and the furthest monitoring station and the houses of the patients was 0.6 and 6 km, respectively (see Fig. S1)."*

- Line 272-273: the REU needs to be calculated at a given value, usually the limit value also set in the Directive. You do not mention those value here, will you use a specific one?

Due to the above-mentioned reasons, we did not to evaluate the REU against a specific value (such as the current limit values). Therefore, we considered the graph displaying all measured concentrations to be a more appropriate option.

In the manuscript we have now added two tables indicating at which $NO_2$ concentration is reached the DQO of 25 % for indoor (Table 4) and outdoor (Table 5).

*Table 4. Concentration in ppb at which the DQO for indicative measurements (25 %) is accomplished for the outdoor calibration.*

| Averaging time | Model | B03-P1 | B03-P3 | B03-P7* | B05-P4 | B06-P4 | B06-P7 | B08-P6 |
|---|---|---|---|---|---|---|---|---|
| 1 min | MLR | N.A. | N.A. | N.A. | N.A. | N.A. | N.A. | N.A. |
|  | SVR | N.A. | N.A. | N.A. | N.A. | N.A. | N.A. | N.A. |
|  | RFR | N.A. | N.A. | N.A. | N.A. | N.A. | N.A. | N.A. |
|  | ANN | N.A. | N.A. | N.A. | N.A. | N.A. | N.A. | N.A. |
| 5 min | MLR | 26 | N.A. | 27 | N.A. | 39 | N.A. | N.A. |
|  | SVR | 23 | 27 | N.A. | N.A. | 34 | 23 | 21 |
|  | RFR | 22 | 28 | 21 | N.A. | 33 | 20 | 21 |
|  | ANN | 24 | 40 | 38 | N.A. | 39 | N.A. | 28 |
| 10 min | MLR | 17 | N.A. | N.A. | N.A. | 29 | N.A. | 28 |
|  | SVR | 17 | 23 | 19 | N.A. | 29 | N.A. | 14 |
|  | RFR | 17 | 22 | 19 | N.A. | 24 | 33 | 17 |
|  | ANN | 17 | 25 | 32 | N.A. | 28 | 38 | 19 |
| 15 min | MLR | 18 | N.A. | N.A. | N.A. | 19 | N.A. | 24 |
|  | SVR | 18 | N.A. | 26 | 35 | 19 | 21 | 11 |
|  | RFR | 17 | 29 | N.A. | 30 | 20 | N.A. | 13 |
|  | ANN | 18 | 44 | N.A. | N.A. | 18 | N.A. | 19 |

*N.A.: not accomplished.*
*\*B03-P7 is an outdoor AQSS used for indoor measurements as part of an experiment to test the outdoor calibration methodology for indoor measurements.*

**Table 5.** *Concentration in ppb at which the DQO for indicative measurements (25 %) is accomplished for the indoor calibration.*

| Averaging time | Model | B01-P4 | B01-P5 | B02-P1 | B02-P6 | B04-P2 | B04-P3 |
|---|---|---|---|---|---|---|---|
| 1 min | MLR | 8 | 11 | N.A. | 42 | 9 | 11 |
| | SVR | 3 | 2 | N.A. | 15 | 2 | 8 |
| | RFR | 1 | - | 40 | 7 | - | 3 |
| | ANN | 5 | 5 | N.A. | 31 | 5 | 7 |
| 5 min | MLR | 7 | 10 | N.A. | 27 | 8 | 9 |
| | SVR | 1 | 14 | N.A. | 15 | - | 6 |
| | RFR | 1 | - | 60 | 21 | - | 2 |
| | ANN | 4 | 4 | N.A. | 22 | 4 | 5 |
| 10 min | MLR | 6 | 10 | N.A. | 26 | 8 | 8 |
| | SVR | 1 | 1 | N.A. | 11 | 1 | 3 |
| | RFR | 1 | - | 39 | 25 | - | 1 |
| | ANN | 3 | 4 | N.A. | 21 | 4 | 5 |
| 15 min | MLR | 7 | 9 | N.A. | 22 | 7 | 7 |
| | SVR | 1 | 4 | N.A. | 4 | 3 | 3 |
| | RFR | 1 | - | N.A. | 20 | - | 3 |
| | ANN | 3 | 4 | N.A. | 19 | 4 | 4 |

*N.A.: not accomplished.*
*The cells marked with (-) do not have a value for the REU as $U_{field}(y_i)$ cannot be calculated with Eq. S6 in the Supplement due to the negative value of $u_s^2(y_i)$ (Eq. S1). This is caused due to the extremely low RSS. Near-zero RSS are an indicator of the overfitting of the RFR in the indoor calibration models.*

A you can see, for some models trained with the data of the indoor calibration methodology, the calculation of the REU using the equations of the CEN/TS 17660-1 (2021) and CEN/TS 17660-2 (2024) gives for RFR calibration models negative values independently from the temporal resolution used. This behaviour has been now explained in the text:

*"The lower REUs that are achieved during the calibration of AQSSs in indoor conditions may be due to the controlled conditions, as the $NO_2$ gas was given stepwise and kept constant for 3.5 hours, as well as the controlled changes of the T and the RH. This lack of variability in the calibration data resulted in low sum of residuals (RSS) triggered by model overfitting."*

*"This creates the challenge of calibrating indoor AQSSs for a wide range of $NO_2$ concentrations and meteorological parameters without causing model overfitting."*

- Line 281: why did you mentioned both directive as the 2024 will overcome the 2008?

The work of WG42 in CEN/TS 17660-1 (2021) and CEN/TS 17660-2 (2024) is aligned with EU Directive 2008/50/EC. However, in this study, we aimed to go a step further by exploring uncertainty in the context of the upcoming regulation. At present, there are no discussions about revising the technical specifications, but this may change in the future.

We have clarified that in the manuscript:

*"The new directive also specifies in Annex V new DQOs for indicative measurements (I.M.) and objective estimation (O.E.) that the Member States shall comply by 11 December 2026. Therefore, the inclusion of new DQOs is intended as forward-looking exercise."*

*"The CEN/TS 17660-1 (2021) and CEN/TS 17660-2 (2024) provides a classification that is consistent with the requirements of DQOs defined in the Directive 2008/50/EC."*

- Figure 3: this kind of figure is not easy to read. I do understand that the main idea is to show a general trend but I'm a bit overwhelmed by so many information.

We acknowledge the difficulty on reading Figure 3 (now Figure 4). We hope the Table 4 and 5 can now help to understand and complement the REU Figures.

- Figure 3: which indicative measurement DQO did you select 2008 or 2024?

As shown in Table 1, the DQO of $NO_2$ for indicative measurements for short-term measurements is the same in both directives (source: Annex I Table A in EU 2008/50/EC and Appendix V, Table 2 in EU 2024/2881).

The manuscript already included the clarification "Note that both directives have the same DQO for indicative measurements (25 %)." We have now added the following clarification in the caption of Figure S3 and S4 in the supplemental material.

*"The DQO for short-term indicative measurements is the same in both Directives."*

- Figure 6: I would advise the author to change the color scheme, currently indoor (pink) and outdoor (dark pink) to get a more distinguishable colors.

We have changed the colours of Figure 6 and Figure 7 (now Figures 7 and 8) to white (indoor) and dark grey (outdoor).

---

## Author Comment (AC2)

**Response to interactive comments from Referee #2**

Thank you for the time you put into reviewing our manuscript and the helpful feedback. Please see our following responses and proposed changes to the original manuscript, which we believe, help to improve this paper and increase its impact. Below the comments from Referee #2 are given in black. Our responses to the comments are shown in blue. Text added or changed in the manuscript is marked in italics.

**General comments**

This manuscript addresses a highly relevant and timely topic: the evaluation of low-cost sensor performance in real-world exposure settings, with a focus on their potential application in epidemiological studies. The proposed approach is valuable, and the work presents several strengths, including the use of a modular sensor platform, the incorporation of multiple performance metrics, and the attempt to link results to regulatory benchmarks. In this regard, the study has the potential to become a significant contribution to the literature on sensors and personal exposure assessment.

We appreciate the reviewer for the positive feedback.

However, the manuscript still requires certain adjustments to reach its full impact. Some elements of the study design are not fully integrated into the discussion of the results, and certain sections would benefit from greater methodological clarity. Overall, the manuscript feels somewhat disorganized, with sections that could be restructured to enhance internal coherence, and the use of technical language could be refined to improve precision. Finally, some conceptual discussions are potentially valuable, but appear somewhat disconnected and would benefit from being more clearly linked to the study's main objectives and findings.

We have taken into account all the given feedback and we believe the new manuscript is now better structured. We have also revised the technical language and the connection between the discussion points and the results.

In summary, this is a manuscript with an original approach and strong potential, which could be significantly strengthened through a comprehensive revision aimed at improving its structure, clarity, and argumentative coherence.

We thank the reviewer for the detailed and insightful review and for the positive statement about our work. We have undertaken a thorough revision of the manuscript with the help of the reviewer´s comments. Specifically, we have:

- Reorganised the sections and subsections to enhance logical flow and narrative consistency.
- Revised the text to improve clarity.
- Clarified the rationale and implications of our findings to strengthen argumentative coherence between the results, discussion and conclusions.

**Comments by section**

**Title**

The current title partially reflects the rich content of the study. In particular, it would be important to highlight that the study does not exclusively address performance, but also includes the calibration process of the sensors, and that its main value lies within the framework of environmental epidemiology A suggestion could be "Calibration and performance evaluation of low-cost PM2.5 and NO2 air quality sensors for environmental epidemiology".

We agree with the reviewer and we have changed the title to the one suggested: "Calibration and performance evaluation of $PM_{2.5}$ and $NO_2$ air quality sensors for environmental epidemiology".

We have just avoided the term "low-cost" in the title because it is bias. Nonetheless, we have added this term as a keyword to help the search engines as it is widely used in the sensor community.

**Abstract**

The abstract effectively summarizes the main elements of the manuscript, but some rewording could improve its clarity and alignment with the content of the paper. For instance, the second sentence ("This study aimed to transcend the limitations...") may come across as somewhat broad and ambitious, and could benefit from a more

specific description of the limitations being addressed. The final sentence of the abstract reads more as an aspirational statement than a conclusion directly supported by the findings. It may be more appropriate to include this type of reflection in the discussion section.

We have removed the wording "transcend the limitations" and the last sentence and added the following sentence in the abstract:

*"Moreover, the impact of the aggregation time (1, 5, 10 and 15 min) on the performance of the calibration models was evaluated for $NO_2$ sensors. Integrating metadata such as activity logs, window status, and data from official monitoring stations, as well as $NO_2$ measurements with diffusion tubes proved to be helpful for data validation and interpretation during the sensor deployment in the houses of the participants."*

**Introduction**
The introduction correctly outlines the distinction between direct and indirect approaches to assessing exposure to air pollutants. However, when discussing the direct approach, the text focuses primarily on mobile personal measurements, which does not accurately represent the methodology used in this study.

We have added more literature about stationary measurements for epidemiological studies.

*"In this context, some studies have evaluated the use of stationary air quality sensors for environmental epidemiology (Morawska et al., 2018; Patton et al., 2022; Bi et al., 2024; Zuidema et al., 2024). Zuidema et al. (2021) evaluated the field calibration based on series of stepwise multiple linear regression calibration models of a low-cost sensor network for multiple gaseous pollutants. They reported the performance achieved using the CV-RMSE and the CV-$R^2$ as well as the limitations of the approach to, for instance, detect the drift of the sensors during deployment or the difficulty to measure low pollution levels. They also discussed about the competing interests forcing the compromise between duration of co-location in order to achieve better calibration (training data) and the deployment for epidemiological purposes."*

Given that a substantial portion of the study focuses on evaluating sensor performance in indoor environments, it would be useful to include in the introduction a more explicit review of previous work in this area. For instance, studies that have assessed the performance of sensors in domestic settings or proposed calibration/validation strategies specifically tailored to indoor conditions. Not including this indoor background limits the reader's ability to grasp from the outset how underdeveloped the literature is in this domain. It also reduces the opportunity to position this study as an important contribution to a still-emerging field, and to contrast the findings with relevant previous work.

We have added more literature about air quality sensor measurements indoors with the focus on how the indoor calibrations were performed.

*"Recent literature demonstrates that stationary indoor air quality measurements with low-cost sensors are widespread, but calibration approaches and durations vary considerably (Anastasiou et al., 2022; Soja et al., 2023; Tryner et al., 2021; Rathbone et al., 2025). Rose et al. (2024) investigated and apportioned the sources of indoor PM at school classrooms using the OPC-N3 (Alphasense, UK). The calibration was carried out using linear regression using data from the co-location with a RI during the exposure to indoor air for 48 h using a time resolution of 1 min. Good agreement for $PM_{2.5}$ (r > 0.85) was reported, without the need of a further correction to account for hygroscopic growth as the RH was below 60 %.*

*Suriano and Penza (2022) tested the performance of Alphasense series B4 sensors for CO, $NO_2$ and $O_3$ during a one-week co-location experiment in a living room using a sampling rate of 2 min. The models tested for calibration were multiple linear regression (MLR), random forest regressor (RFR), artificial neural networks (ANN) and support vector regressor (SVR), and the input parameters used were the working electrode (WE), the auxiliary electrode (AE), the temperature (T) and the RH or also the net difference WE – AE, including also the T and RH. They proved that the $NO_2$ measurements were in good agreement ($R^2 > 0.7$, 8.4<MAE< 12 ppb, 10.6 < RMSE< 16.3 ppb) if calibrated through MLR, RF and ANN, having the best results when using separately the sensor electrode signals as inputs. Note that in both studies the co-location was short, as the pumps of the RI are too noisy to keep the instrument longer periods in such indoor environments."*

One of the most valuable aspects of the manuscript is the use of regulatory metrics, such as the calculation of relative expanded uncertainty (REU) as defined in European directives, to assess sensor performance. However,

these directives, and the related technical specifications, were not designed for personal monitoring. This represents an opportunity for the manuscript to elaborate more thoroughly on an important gap in the literature: the relevance, limitations, and potential of using regulatory standards as quality criteria in epidemiological studies. Addressing this point would strengthen the conceptual justification of the chosen approach and connect it to recent debates on the adaptation of regulatory frameworks to emerging technologies.

We have added a new explanatory figure (Figure 1) and the following text explaining the added value of this paper:

*"As shown in the aforementioned examples from the literature, it is common practice to report sensor accuracy primarily through metrics such as R² or Pearson correlation coefficient, with some studies including additional statistics like MAE or RMSE. However, these statistical evaluations alone may not be sufficient for specific purposes as well as for stakeholders such as environmental agencies, who work with expensive instrumentation that undergo rigorous calibration and continuous performance assessments throughout their operational lifespan (Flores et al., 2012; Flores et al., 2013). Therefore, to build greater confidence in air quality sensor data, more comprehensive validation protocols and calibration procedures are essential.*

*Figure 1 shows the link between epidemiological studies, the WHO and the European Directives for air quality. The epidemiological studies are the prove of causality between air pollutant exposure and health effects, and they are reviewed by the WHO to recommend the limit values which are the guidance to set the air quality regulations. In the European Union, the EU Directive 2008/50/EC and the new Directive 2024/2881 specify the short- and long-term limit values, as well as the Data Quality Objectives (DQOs) that the measurements must meet for ambient air quality assessment, depending on the type of measurement (fixed, indicative or objective estimation). At this moment there is no DQOs for indoor air quality assessment. However, in this work we have evaluated the sensor data for the indicative and objective estimation DQOs set in the directives for both indoor and outdoor measurements."*

[Figure]

***Figure 1.*** *Interconnection between epidemiological studies, the WHO air quality guidelines, and the DQOs established in the EU Air Quality Directives.*

**Methodology**
Section 2.1 includes valuable elements of the study design, such as pulmonary function tests, health surveys. However, these elements are not developed or connected with the results presented. It would be useful to clarify whether these data fall outside the scope of this manuscript or if they will be analyzed in future work. Otherwise, the reader may perceive a disconnect between the protocol and the actual focus of the study.

We have included a citation to our publication in which we evaluate this information in detail.

*"This information collected from each participant has been further analysed in Chacón-Mateos et al. (2024)."*

The description of sensor deployment would benefit from more detailed information on installation conditions, such as the height above ground, orientation and distance from potential sources (e.g., windows, heaters, traffic), and the ventilation characteristics (passive or active?) of the protective enclosure.

The potential sources nearby were recorded in the environmental questionnaire at the first visit. More information about it was published in the supplemental material of my publication "Feasibility Study on the Use of $NO_2$ and $PM_{2.5}$ Sensors for Exposure Assessment and Indoor Source Apportionment at Fixed Locations".

We have added in the manuscript the following information about the sensor installation conditions:

*"Both AQSSs had a passive ventilation system."*

*"The indoor AQSSs were installed in the participants' living rooms, as this space was identified as the primary area for their daily activities. The exact placement within the living room was determined by the proximity to a power outlet and the availability of suitable space, with devices most commonly positioned on a table or TV stand. Outdoor AQSSs were installed in a variety of locations, including hanging from balconies, placed on window sills, or positioned on terrace floors, with placement always dependent on the availability of a power socket. A summary of the information collected in the environmental questionnaire about the neighbourhood, the building as well as the home environment, including the type of windows, possible pollutant sources indoors and outdoors, can be read in Chacón-Mateos et al. (2024)."*

The title "Study design and quality assurance" could be revised, as subsections 2.3.1 and 2.3.2 focus on quality control rather than experimental design.

We have changed the title of section 2.3 to *"Quality assurance".*

The title of subsection 2.3.1, "Sensor calibration before deployment," also seems inappropriate (same with the title of Table 1), since the content primarily addresses the co-location experiments necessary for calibration but does not include the full calibration process, which is described in other sections (e.g., 2.3 Sensor correction models). Reorganizing these subsections could improve clarity and support reader comprehension.

We have changed the title of the subsection to *"Sensor co-location before deployment"*. We have also changed the caption of Table 1 to *"Co-location methodology of the $NO_2$ and $PM_{2.5}$ sensors."*

Moreover, the section "Sensor correction models" is now called "Calibration procedures" and has been added as a subsection of section 2.3 "Quality Assurance".

Some passages in subsection 2.3.1 could be edited for clarity, as they currently lack precision. For example, statements like "The experiments in the particle chamber took about an hour and were repeated at least twice" or "the sensors were co-located in the laboratory for some days" are somewhat vague.

We have made several changes in the whole section to improve the precision of the text. For example, we have changed the word "calibration" for "co-location" where it was needed. The following sentences were also improved:

*"The experiments in the particle chamber took one hour and were repeated twice. … After the co-location in the particle chamber, the sensors were installed in the laboratory for a minimum of 2 days and a maximum of 27 days to expose the sensors to real PM indoors."*

Since the selected co-location site (Hauptstätter Straße) is a known urban hotspot with elevated pollution levels, a brief methodological justification for this choice would be useful.

We have added the following justification:

*"The outdoor AQSSs were co-located seven to 34 days in the hotspot monitoring station at Hauptstätter Street (48°45´55.8936´´ N, 9°10´12.9396´´ E), that belongs to the University of Stuttgart. The average co-location time was 15 days. The advantage of performing the co-location in a hotspot station is that the maximum concentrations expected in the city are covered. However, low-concentrations are unusual to happen there and that may cause a lack of low-concentrations for the training data."*

Throughout the manuscript, the term "reference instruments" (RIs) is used generically, but under European legislation there are two levels of "fixed measurements": reference instruments and equivalent to reference. It would be helpful to specify which category applies to each instrument used in this study.

We have changed the meaning of RI to reference-equivalent instrument.

Further clarification is also recommended in section 2.3.2: Which sensors were validated against which official stations? How far apart were they? How was agreement with the passive diffusion tubes quantified? What interpretive criteria were applied for indoor PM2.5 records? Table 2 indicates that the validation with diffusion tubes was quantitative, how was the quality of those measurements ensured?

Only the outdoor sensors for $PM_{2.5}$ and $NO_2$ were compared with the data of the monitoring stations. As we collected all the data, in a first analysis, all of them were used. For the Figures of the manuscript we have just selected one either due to the similarity in the type of area (hotspot, urban) between the sensor location and the monitoring station or due to the shortest distance.

Regarding the diffusion tubes, the agreement of the sensor data with the diffusion tubes was quantified by comparing the values of $NO_2$ measured with the diffusion tubes during 14 days to the average of the continuous sensor data using different calibration models during those 14 days. Moreover, in order to ensure the quality of the measurements carried out with the diffusion tubes, we co-located three diffusion tubes (triple determination) in the monitoring station of the University of Stuttgart and changed them every 14 days. We have also included the citation of the standard EN 16339: Ambient air - Method for the determination of the concentration of nitrogen dioxide by diffusive sampling.

Due to the lack of passive samplers for $PM_{2.5}$, the indoor $PM_{2.5}$ levels could only be validated using the activity logbook—by checking whether peak concentrations coincided with activities likely to generate particulate matter (e.g., cooking, cleaning, grilling), or by analysing window status (open/closed) and temperature variations.

We have included the following text in the manuscript:

*"The agreement or disagreement of the sensor data with the diffusion tubes was quantified by comparing the values of $NO_2$ measured with the diffusion tubes during 14 days to the average of the continuous sensor data using different calibration models during those 14 days. For patients P2 and P4, only one period was collected of 14 and 19 days, respectively. Moreover, in order to ensure the quality of the measurements carried out with the diffusion tubes, we co-located three diffusion tubes (triple determination) in the monitoring station of the University of Stuttgart and changed them every 14 days."*

*"Additionally, the data of the four outdoor air quality monitoring stations available in Stuttgart as well as the data of the monitoring station from the University of Stuttgart in Hauptstätter Street was also collected to qualitatively compare their $NO_2$ and the $PM_{2.5}$ trends with the data of the outdoor AQSS during deployment in the houses of the patients. The air distance between the closest and the furthest monitoring station from the houses of the patients was 0.6 and 6 km, respectively (see Fig. S1)."*

*"Due to the lack of passive samplers for $PM_{2.5}$, the indoor $PM_{2.5}$ concentrations could only be validated using the activity logbook, by checking whether peak concentrations coincided with activities likely to generate particulate matter (e.g., cooking, cleaning), or by analysing window status (open/closed) and temperature variations."*

The title of section "2.3 Sensor correction models" does not accurately reflect the content, as the methods described (MLR, SVR, RFR, ANN) are in fact calibration procedures. The manuscript uses the terms "correction" and "calibration" somewhat interchangeably. Since the study has a strong methodological focus on performance evaluation and the application of statistical calibration models, more precise terminology is recommended. I suggest referring to the International Vocabulary of Metrology (https://www.bipm.org/documents/20126/2071204/JCGM_200_2012.pdf). It would also be helpful to include a summary table that, for each pollutant and environment combination, specifies: the model used, the explanatory variables, the reference instrument type, and details on training, testing, and validation. What temporal resolution was used to train the models?

We have changed the title of section 2.3 to "Calibration procedures" and we have included it as a subsection of section 2.3 "Quality Assurance".

We have changed the word "correction" for "calibration" in the sentences where "calibration" makes more sense according to the International Vocabulary of Metrology JCGM 200:2012 document.

The information suggested to be included in a summary table was included in the supplemental material for $PM_{2.5}$ sensors. For the $NO_2$, we prefer not to include the summary table because it would be too big (7 patients, 2 sensor systems per patient (indoor, outdoor), four calibration models per $NO_2$ sensor (MLR, RFR, SVR, ANN)

and four different temporal resolutions per calibration model (1, 5, 10, and 15 min). Details about the training, testing are already included in Table 1 and 2.

Section 2.4 indicates that the models were trained using 15-minute aggregated data, while section 2.5 reports performance evaluated at 1, 5, and 10 minutes. It is important to clarify whether models trained at 15-minute resolution were directly applied to data at other resolutions, and what the limitations of this approach are.

Following the previous recommendations, we have changed the title of this section to "Performance evaluation" and added as a subsection of 2.3 "Quality assurance". Moreover, the section "Data pre-processing" has also been added as part of section 2.4 "Quality assurance" and the title has been changed to "Data processing".

In Section 2.4, (now 2.3.4) we have modified the following sentence for clarity:

*"In the case of the calibration of the $NO_2$ sensors, we evaluated the impact of the averaging time in the model performance by using 1-, 5-, 10- and 15-minute averages for both training and testing datasets".*

In section 2.5, where the use of target diagrams is described, a more stringent performance criterion is defined using a threshold of 0.5. However, this threshold appears arbitrary. It would be helpful to justify its selection, or to clarify that it is an exploratory criterion adopted for this study.

The performance goal was inspired by previous studies (Thunis et al., 2012; Jolliff et al., 2009; Bagkis et al., 2021). We have included the mentioned references and clarified the selection in the manuscript with the following sentence:

"This threshold is an exploratory criterion adopted specifically for the purposes of this study."

This section combines references to technical standards CEN/TS 17660-1 (2021) and 17660-2 (2025), which are specific to sensors, with DQOs from EU directives 2008/50/EC and 2024/2881, which were not designed for low-cost devices or indoor measurements. This could mislead readers (especially those unfamiliar with EU regulation) into thinking that sensors are expected to comply with the stricter DQOs from the 2024/2881 directive, which is not the case in current practice or within the scope of existing CEN/TS guidance. It is also important to note that both the CEN/TS standards and the cited directives are aimed at outdoor ambient air quality and set minimum temporal resolutions of one hour or longer (depending on the pollutant). Therefore, their direct application to shorter averaging intervals, like those used in this study, is not formally covered. It is suggested to explicitly state whether the inclusion of new DQOs is intended as an exploratory or forward-looking exercise, and to clarify that the current regulatory framework remains anchored in the 2008/50/EC directive and its associated DQOs. Simplifying the language in this section could also help improve accessibility for non-specialist readers.

In Section B of Annex VI of the EU Directive 2024/2881 it can be read the following text:

"A Member State may use any other method which it can demonstrate gives results equivalent to any of the reference methods referred to in Point A of this Annex or, in the case of particulate matter, any other method which the Member State concerned can demonstrate displays a consistent relationship to the reference method, such as an automatic measurement method that meets the requirements in standard EN 16450:2017 'Ambient air – Automated measuring systems for the measurement of the concentration of particulate matter ($PM_{10}$; $PM_{2.5}$)'. In that event, the results achieved by such other method shall be corrected to produce results equivalent to those that would have been achieved by using the reference method."

As I understand it (and please correct me if I'm wrong), the EU Directive leaves the door open to air quality devices that use alternative measurement principles, provided the method can deliver equivalent results to the reference method. Moreover, in the scope section of both CEN/TS 17660-1:2021 and 17660-2:2024 it can be read: "This document provides a classification that is consistent with the requirements for indicative measurements and objective estimation defined in Directive 2008/50/EC." That means that it is actually expected that some sensors may fulfil the requirements for being used as indicative measurements or objective estimations. That is also the reason why in the manuscript we have omitted the DQO of fixed measurements ans we have focused on the DQO for indicative measurements and objective estimations.

I see the point regarding indoor measurements and the temporal resolution. We have included the following text:

*"At this moment there is no DQOs for indoor air quality assessment. However, in this work we have evaluated the sensor data for the indicative and objective estimation DQOs set in the directives for both indoor and outdoor measurements."*

*"Note that the short-term DQOs were conceived for hourly and daily averages for $NO_2$ and $PM_{2.5}$, respectively."*

We have also modified these sentences for clarification:

*"The new directive also specifies in Annex V new DQOs for indicative measurements (I.M.) and objective estimation (O.E.) that the Member States shall comply by 11 December 2026. Therefore, the inclusion of new DQOs is intended as forward-looking exercise."*

*"The CEN/TS 17660-1 and CEN/TS 17660-2 provides a classification that is consistent with the requirements of DQOs defined in Directive 2008/50/EC."*

The statement in line 341 regarding the need for "enough data points" to train the models is relevant but not further developed in the text. It would be valuable to include a more explicit discussion of the actual number of data points used in this study for calibrating each model (especially the NO2 models based on machine learning), and whether that amount is adequate given the complexity of the algorithms. Additionally, the expression "low uncertainties" may be ambiguous to some readers, as it refers here to the uncertainty associated with the calibration model (i.e., how well the functional relationship between raw and calibrated data reflects the actual relationship), and not to total measurement uncertainty (of which calibration uncertainty is only one component). Clarifying this distinction would help avoid potential misinterpretation.

There is an explicit discussion about the number of data points and their effect of the training of machine learning algorithms in section 4.1.3. We have changed Figure S7 (now Figure S5) of the Supplemental materials to a boxplot graph where more information about the number of training data points per temporal resolution is displayed (before was only the average). Moreover, we have removed the expression "low uncertainties" and added the following text in the results:

*"In our study, we found 10-minute average time a good compromise between training the models with enough data points and reaching the DQO for indicative measurements at an average of 23 ppb for the outdoor $NO_2$ sensors. In the Fig. S5 of the Supplement, the number of data points for the training of the $NO_2$ calibration models for the different time resolutions is shown for the indoor and outdoor sensors. A detailed study about the effect of eleven temporal resolutions (between 10 s and 6 h) in the performance of $NO_2$ sensors can be read in Schmitz et al. (2025)."*

**Results**
In section 3.1.1, while PM2.5 was modeled using a linear approach with modest data volumes (200-600 data points per sensor), the NO2 models rely on machine learning algorithms (MLR, SVR, RFR, ANN), which typically require significantly larger datasets to avoid overfitting and ensure model stability. Since no table equivalent to Table S5 is provided for NO2, and the data volumes used are not explicitly reported, it is difficult to assess the validity and robustness of the applied models. See also the related comments on section 2.5.

The new Figure S5 in the Supplement displays in a clearer way the number of data points used for the training of the $NO_2$ calibration models.

In section 3.1.2, all NO2 calibration models fall on the left-hand side of the target diagram. It would be relevant to consider whether this reflects a systematic underestimation of the actual variability by the calibrated sensors, or a limitation in the dynamic response range of the devices.

If all the results of the calibration models are in the left quadrants, that means that their standard deviation is lower than the standard deviation of the RI (Zimmerman et al., 2018). This can indeed reflect a systematic underestimation of the actual variability by the calibration models. We have added the following sentence:

*"This could reflect a systematic underestimation of the actual variability by the calibration models."*

The inclusion of multiple additional performance metrics in this section (MAE, MEF, slope/intercept, r) does not appear to be clearly justified, especially given that robust tools such as REU and target diagrams (which integrate error and bias) are already applied. Without a clear explanation of their added value (particularly from

the perspective of potential users such as researchers, agencies, or communities) these metrics may come across as a statistical exercise without a defined purpose. Moreover, their inclusion could obscure the central message of the manuscript. It is recommended to focus this section on the key metrics relevant for practical decision-making and consider moving the rest to the Supplement.

We agree that REU and the target diagrams are more robust tools than the performance metrics (and therefore the chosen order of the sections in the Section "Results"). However, the use of performance metrics is much more common in the literature than the use of REUs and target diagrams. As a consequence, comparing results with other studies is almost impossible without the inclusion of this metrics. Therefore, we would like to retain the performance metrics as part of the main manuscript.

We have restructured the paragraphs in the subsection 2.3.5 "Performance evaluation" to start with the REUs, followed by the explanation of the target diagrams and concluding with the statistical metrics. We have also included the follow explanation of the added value of including the performance metrics:

"By presenting both conventional performance metrics and more robust diagnostic tools, we aim to enable a broader comparison with other studies as the REUs and target diagram are still scarcely used in the performance evaluation of AQSSs."

In section 3.2.1, passive samplers are used to validate sensor performance; however, no information is provided about the quality/uncertainty of the data they produce.

We have added the error bars indicating the uncertainty of the passive samples in Figure 9 and we have clarified the meaning of the error bars in the caption with the following text:

*"Error bars indicate the expanded uncertainty of the diffusion tubes (18.4 %)."*

Section 3.2.3 presents the use of activity logs as a strategy to contextualize sensor data and explore individual exposure patterns. It is suggested to frame this as an exploratory or complementary tool, and to consider whether previously cited studies offer methodological frameworks that could support a more systematic integration of metadata in future research on personal exposure.

We have included the following text:

*"In this section, we present an example of how the use of metadata, specifically the activity and window status log, can be used as a complementary tool to validate and understand sensor data in places without RI."*

*"Other studies, such as that by Novak et al., have proposed methodological frameworks that more systematically integrate metadata from activity logs with air quality sensor data (Novak et al., 2024; Novak et al., 2023a; Novak et al., 2023b)."*

**Discussion**
The final paragraph of section 4.1.1 introduces the "EASE" methodology, but it is unclear what exactly the authors are referring to, which hinders its interpretation. Additionally, this paragraph includes some generalizations that do not appear to be directly supported by the study's results. It is recommended to revise the paragraph to more accurately reflect the exploratory nature of this reflection.

We have modified the text and now it reads as follows:

*"Other solutions for indoor sensor calibration could be a hybrid calibration like the Enhanced Ambient Sensing Environment (EASE), which combines the advantages of laboratory calibration with the increased accuracy of field calibration (Russell et al., 2022). To date, this approach has only been tested with multilinear regression models and in outdoor environments. Further research is needed to determine whether it is suitable for indoor environments and the training of machine learning algorithms."*

Section 4.1.2 would benefit from being restructured to clearly distinguish between: (i) the review of previous studies, (ii) the authors' own findings, and (iii) the methodological decisions taken. It would also be useful to discuss whether the use of a linear model to incorporate ozone was the most appropriate choice, considering that sensor cross-sensitivity to O3 is not necessarily linear or constant.

We have divided the text in Section 4.1.2 into two paragraphs: one focusing on the comparison of our results with previous literature, and the other discussing the test involving ozone data.

Rather to discuss about the use or a linear regression to incorporate ozone data into the calibration model, I would like to draw the attention in the consequences that adding external data from a neighboring measurement station could have in the integrity of the model.

*"Moreover, we would like to highlight that incorporating data from a neighbouring station as an input feature for training the calibration models was identified as a 'questionable parameter' by Hagler et al. (2018), as it may compromise data integrity, blurring the line between an actual measurement and a model prediction (Hayward et al., 2024)."*

Section 4.1.3 addresses the effect of averaging time on $NO_2$ sensor calibration, but the empirical evidence is limited due to the short duration of the calibration phase (two weeks). It is recommended to report how many data points were used for each averaging interval (1, 5, 10, 15 min) and to discuss how variability in sample size affects the comparison. The statement "Our study demonstrated…" could be softened, as the available data do not support a generalizable conclusion. Moreover, the reduction in MAE is not contextualized in relation to other performance metrics, nor is its practical relevance evaluated, which makes it difficult to interpret the actual benefit of using a 10-minute average.

We have added more information about the number of data points for training and modified the mentioned sentence:

*"The choice of the temporal resolution significantly affects the quality of training data for $NO_2$ sensor calibration models. Even if the number of data points using a temporal resolution of 1 min was between 28,000 and 5,100 for indoor calibrations and between 14,400 and 5,500 for outdoor calibrations (see Figure. S5), these high-resolution data contained more noise, which impacted negatively the training quality."*

*"Our study found that using a 10-min averaging period over a two-week calibration phase (comprising between 400 and 2,500 data points) resulted in a lower MAE for $NO_2$ sensors. However, the difference compared to a 15-minute averaging period was small across most metrics, including target diagrams, REUs and the comparison with the $NO_2$ concentrations measured by the diffusion tubes."*

One of the recommendations in section 4.3 states: "we recommend adjusting the averaging time based on the available data...". This may be confusing or lead to methodologically questionable interpretations. The choice of temporal resolution should primarily be guided by the study's objectives (e.g., capturing acute exposure events), not just by the amount of data available. It is suggested to rephrase this recommendation or to justify its applicability in more detail.

We understand that this may be confusing; therefore, we have removed the sentence.

The closing paragraph of this section suggests that, due to the poor performance of sensors at low concentrations, they may be more suitable for environments with high pollution levels. However, this statement is not supported by empirical evidence in the present study. It is recommended to revise this conclusion to more directly acknowledge the limitation observed, rather than inferring a potential use case that was not investigated here.

We have modified the paragraph as follows:

*"In summary, while the tested sensor units generally fulfil the DQOs for higher concentrations, the higher REU of the sensors at lower concentrations may hinders their application in epidemiological studies. Despite limitations at low pollutant levels, calibrated AQSSs are a promising tool to increase the ubiquity of epidemiological studies for low- and middle-income countries or regions where higher air pollutant concentrations are expected, where more epidemiological studies are needed (Amegah, 2018)."*

The reflection in section 4.4 on the meaning of the term "low-cost" is valid, but feels disconnected from the rest of the manuscript. If this content is to be retained, it is suggested to move it to the conclusions and, most importantly, to explicitly connect it to the study's findings. For instance, the authors could discuss whether the calibration, validation, and performance requirements observed in this study still justify labeling these sensors as "low-cost" in real-world applications. Otherwise, the section may come across as tangential or anecdotal.

We agree that this paragraph feel disconnected. However, their inclusion in the conclusions would not help as it is not related to the objectives. It is rather a lesson learned through the study. We have made some modification focusing on their connection to epidemiological studies so we hope can stay as a separate subsection in the discussion. We have divided the text into two paragraphs and included the following sentence:

*"Moreover, the use of AQSSs in health studies requires the acquisition of RI for their calibration, as well as additional time for co-location, which must be accounted during the planning phase."*

**Conclusions**
The conclusions section brings together various elements of the study, but does not clearly articulate a central message regarding the value, applicability, and limitations of the evaluated sensors for epidemiological research. It is recommended to revise this section to: (i) avoid generalized claims that are not supported by the evidence presented, (ii) strengthen the critical analysis of the results (e.g., the limited transferability of laboratory calibrations, or the actual utility of machine learning models with small datasets), and (iii) better guide the reader regarding practical implications and future directions. Reformulating the conclusions based on the study's initial objectives would help close the manuscript with greater clarity and strength.

We have modified the conclusion taken into account the suggested revisions:

*"In this study, we evaluated the performance of the OPC-R1 and the B43F sensor models for measuring $PM_{2.5}$ and $NO_2$, respectively, for their use in health studies across both indoor and outdoor microenvironments. For that purpose, we used REUs, target diagrams and common error metrics. A central research question concerned whether calibrated sensors could meet the DQOs defined in the EU Directive 2008/50/EC and in the recently published EU Directive 2024/2881, and if so, at which concentration levels.*

*The co-location phase was conducted two weeks before the deployment, where the data from RI were used to calibrate the $PM_{2.5}$ sensors with ULR and test regression (MLR) and ML models (RFR, SVR and ANN) to calibrate the $NO_2$ sensors. The results show that the REUs depend on the temporal average (i.e. the number of data points) used during the training. Generally, coarser averaging times (10- and 15-min) improved the likelihood of meeting the DQO at lower concentrations while high-resolutions (1- and 5-min) led to higher REUs due to the impact of the sensor noise in the training data.*

*The validation of the sensor data during deployment in the houses of the patients was performed using $NO_2$ diffusion tubes, patient logbooks with activity information and window status as well as data from the monitoring stations in Stuttgart. Even though ML seems a promising tool in the field of AQS, the training data acquired by exposing the sensor and the co-located RI to artificially generated $NO_2$ for indoor calibration did not yield realistic results (compared to the $NO_2$ measurements of the diffusion tubes) for some of the ML models (RFR and SVR). Furthermore, performance evaluation revealed that calibrating PM sensors using liquid paraffin as a test aerosol is problematic, owing to mismatches between the assumed particle density in the sensor's internal algorithm and the actual density of the generated aerosol.*

*Our results highlight that the environmental conditions (e.g. temperature and relative humidity ranges) and concentration levels present in the training phase are critical for ensuring reliable data when sensors are relocated. The choice of temporal averaging used to train the models directly affects the range of concentrations, temperatures and RH covered and, consequently, it has a direct impact on the performance of the calibration model. Moreover, the integration of metadata, such as activity logs, window status, data from official monitoring stations and diffusive samples, was proved a good tool for validating and interpreting sensor data.*

*There remains a need for more comprehensive sensor evaluations that extend beyond basic statistical metrics such as $R^2$ and MAE. Tools like REUs and target diagrams add significant value by enhancing trust and transparency in sensor data. Future work should also prioritise assessing the transferability of calibration models, particularly those developed in indoor co-location settings, to enable the integration of reliable and traceable air quality sensor data in future health studies."*

**Specific comments**
Line 29: Consider adding "fine fraction" when referring to PM2.5.

It has been added.

Lines 45-46: Suggest adding a reference to support the statement.

Two new citations have been added (Yun and Licina, 2023; Bendl et al., 2023).

Line 62: It would be useful to acknowledge that, although the high temporal resolution offered by sensors is valuable for capturing dynamic exposures, it also increases instrumental noise, which directly impacts measurement uncertainty, particularly important in epidemiological studies.

*It has been added and we have included a new citation (Schmitz et al., 2025).*

Line 82: It may be appropriate to introduce that this is a study attempting to apply a direct measurement approach.

*The study applies an indirect approach in terms of exposure quantification (direct would be personal monitoring using wearables). We prefer not to change the sentence.*

Lines 88-89: Consider rephrasing for improved clarity.

*The sentence has been rephrased:*

*"A univariate linear regression (ULR) calibration was investigated to correct the $PM_{2.5}$ sensor measurements. The outdoor $PM_{2.5}$ sensor included a thermal drying inlet."*

Lines 95-96 and 103-104 could be rewritten for a more natural and concise expression.

*The sentences have been rephrased:*

*"One participant agreed to install two outdoor air quality sensor systems (AQSSs) instead of one, to compare street-side and garden-side concentrations."*

*"Participants also completed a spirometry test, a health survey on their symptoms, and a logbook documenting hourly indoor activities, window status and presence at home."*

In section 2.2 (and in other parts of the manuscript), the terms "air quality monitor", "sensor system" and "AQSS" are used interchangeably without clarifying whether they are synonymous.

*We have changed the term "air quality monitor" and "sensor system" for "AQSS".*

Check the numbering of the "Study design and quality assurance" section (2.3?).

*It has been corrected.*

In Table 1, clarify what is meant by "low concentrations".

*It has been clarified (< 10 ppb).*

Lines 161-162: Clarify what the authors are referring to.

*The sentence is has been clarified:*

*"To account for potential particle losses caused by electrostatic forces from the plastic enclosure, the entire indoor AQSS was placed inside the chamber."*

Lines 249-250: The wording could be improved for clarity and precision.

*The sentence has been clarified:*

*"The warm-up period of $NO_2$ sensors was observed to range from four hours up to three days and was manually removed after visual inspection of the data."*

Line 253: This statement could be reconsidered or qualified: "In theory, the higher the number of data points, the better the model performs", as model performance does not necessarily with more data points.

*We have removed it and clarified the following sentence:*

*"Note that the $NO_2$ sensor signal exhibited significant noise, making necessary to balance the number of training data points with effective noise reduction in order to optimise model performance."*

Lines 270-271: "Higher performance" could be replaced with "better performance" or a more neutral formulation. Also, "in general" suggests there may be exceptions, but that's not applicable here, where the interpretation of target diagrams is systematic.

We have removed "in general" and changed "higher" for "better".

Line 287: "Resolution" appears twice.

We have re-written the sentence:

"*For epidemiological studies, however, especially those using portable monitors, 24-h average or even 1-h average may be insufficient, as detecting short-term pollution peaks requires higher temporal resolutions.*"

Line 288: The phrase "especially in mobile measurements" seems out of place in the context of this study, since no mobile measurements were performed. In the following sentence, there may be a typo ("long" instead of "longer"?).

We have removed the mention to the mobile measurements and changed "long" for "longer".

Consider including a brief explanation of the nomenclature used for sensor IDs.

We have modified some sentences to make the selection of the sensor ID more intuitive:

"*The indoor AQSSs (B01, B02 and B04…*"

"*Outdoor AQSS units (B03, B05, B06 and B08)*"

"*…for patient P7 an outdoor box (B03-P7) calibrated…*"

Figures are in general a bit difficult to interpret, are different models, averaging intervals, or measurement phases being shown? It is suggested to include a complete legend in each figure or, alternatively, clearly describe in the text what each marker represents.

We have added more information in the captions:

"***Figure 3.*** *REU for (a) indoor and (b) outdoor PM$_{2.5}$ sensors. The coloured symbols are different AQSSs which were deployed in the house of different patients (B0X-PX) and therefore some AQSSs were through more than one calibration phase. B03 was calibrated before the deployment in the house of patient 7 (B03-P7-start) and after the deployment (B03-P7-end). The dashed lines indicate the DQOs for indicative measurements while the dash-dot lines represent the DQOs for objective estimation (black for EU Directive 2008/50/EC and red for EU Directive 2024/2881).*"

"***Figure 4.*** *Example of REU for (a) indoor and (b) outdoor NO$_2$ sensors for the tested models (in different colours MLR, SVR, RFR and ANN) at different averaging times (in different symbols 1 min, 5 min, 10 min and 15 min) against reference concentrations. The dashed line indicates the DQO for indicative measurements while the dash-dot lines represent the DQOs for objective estimation (black for EU Directive 2008/50/EC and red for EU Directive 2024/2881). The DQO for short-term indicative measurements is the same in both Directives.*"

"***Figure 5.*** *Target diagrams for (a) indoor and (b) outdoor PM$_{2.5}$ sensors. The coloured symbols are different AQSSs which were later deployed in the house of different patients (B0X-PX) and therefore some AQSSs were through more than one calibration phase. B03 was calibrated before the deployment in the house of patient 7 (B03-P7-start) and after the deployment (B03-P7-end).*"

"***Figure 6.*** *Example of target diagrams for indoor (left) and outdoor (right) NO$_2$ sensors for the tested models (in different colours MLR, SVR, RFR and ANN) and different averaging times (in different symbols 1 min, 5 min, 10 min and 15 min).*"

"***Figure 9.*** *Comparison of the NO$_2$ models with the concentration measured by the passive samples (two-week period) for (a) indoor and (b) outdoor sensors. Models (in different colours) were trained with data averaged every 10 min. Error bars indicate the expanded uncertainty of the diffusion tubes (18.4 %).*"

"***Figure 10.*** *Time series of hourly outdoor (a) NO$_2$ calculated using the four tested calibration models (in different colours MLR, RFR, SVR and ANN) and (b) PM$_{2.5}$ concentrations calculated using the ULR calibration*"

*model during deployment in the house of patient P1 as well as the RH. The data of the RI is shown with a solid black line in both graphs."*

Figure 8 does not show results for participants P1 and P3, and the text in section 3.2.1 does not explain their exclusion. The sampling period (14 or 15 days, as mentioned in the text) is also not indicated. Recommend expanding or clarifying.

We have added an explanation to that:

*"No diffusion tubes could be installed in the house of patient P1 due to a delay in the delivery. No outdoor data in the house of patient 2 is shown as it was lost due to a storm."*

The sampling period is written in the caption as "two-week period".

The term "passive samples" is used throughout the text. A less ambiguous expression could be "passive sampler measurements" or "passive sampling data".

We have changed "passive samples" for "diffusion tubes".

Line 455: It seems to refer to the data collected by the monitoring station, not the station itself. Check for clarity.

It has been corrected.

Line 456: It is recommended to specify more clearly that the sensor was located outdoors, outside the second-floor window, to avoid ambiguity.

We have included the word "outdoor".

Lines 464-465: It is stated that this is a "clear example" of the effect of temperature and relative humidity on electrochemical sensors. However, the presented data do not allow us to determine whether the observed error is due to sensor limitations or extrapolation of the model beyond the training range.

The sentence has been removed.

In Figure 9b (PM2.5), relative humidity is included as a contextual variable, which may be misleading since, according to the methodology, RH was not used as a variable in the PM calibration model. Given that RH was used for NO2 models (Figure 9a), this graphic choice could be misleading and would benefit from clarification.

The addition of the RH to the Figure 9 (b) (now Figure 10) help to understand the effect of the low-cost dryer, which is mentioned in the text, even though it is not the focus on this manuscript as the evaluation of the low-cost dryer was already published (AMT - Evaluation of a low-cost dryer for a low-cost optical particle counter). We believe that some readers will appreciate having the data of the RH in the Figure 9 (b). As for Figure 9 (a), yes, it is true that the RH was used for the calibration models. The decision not to display it is purely aesthetic, as adding more lines would overcrowd an already busy graph. We have, however, included the lines for T and RH in the Figures S8 – S12 in the supplemental material for those readers who are willing to spend more time understanding and evaluating the effect of this variables in the calibration models.

Line 470: The statement that machine learning models "consistently overestimated" concentrations compared to passive tubes may be overstated, as only one biweekly average value per participant is available, and the comparison is limited to a few cases (Figure 8).

The text has been modified:

*"Although the calibration models tended to overestimate concentrations compared to the diffusion tubes, the SVR and RFR models did not exhibit overfitting, unlike what was observed with the indoor calibration. Note that in Stuttgart outdoor $NO_2$ concentrations are generally higher than indoor concentration, and the models are not adept at extrapolating to lower concentrations. That represents a challenge for using an outdoor co-location to calibrate a $NO_2$ sensor for indoor measurements."*

The use of the term "addition of ozone" in lines 531 and 535 could be misleading, as in the context of atmospheric chemistry or sensor testing it is often interpreted as the physical addition of ozone gas to an experimental mixture.

We have added the word "data" to "ozone".

**References**

[revised manuscript text omitted]